# Qki regulates myelinogenesis through Srebp2-dependent cholesterol biosynthesis

Xin Zhou[1,2†], Seula Shin[1,3†], Chenxi He[4†], Qiang Zhang[1], Matthew N Rasband[5], Jiangong Ren[1], Congxin Dai[1,6], Rocío I Zorrilla-Veloz[1,3], Takashi Shingu[1], Liang Yuan[1,7], Yunfei Wang[8], Yiwen Chen[9], Fei Lan[4], Jian Hu[1,3,10]*

[1]Department of Cancer Biology, The University of Texas MD Anderson Cancer Center, Houston, United States; [2]Cancer Research Institute of Jilin University, The First Hospital of Jilin University, Changchun, Jilin, China; [3]Cancer Biology Program, MD Anderson Cancer Center UTHealth Graduate School of Biomedical Sciences, Houston, United States; [4]Shanghai Key Laboratory of Medical Epigenetics, International Co-laboratory of Medical Epigenetics and Metabolism, Ministry of Science and Technology, Institutes of Biomedical Sciences, Fudan University, and Key Laboratory of Carcinogenesis and Cancer Invasion, Ministry of Education, Liver Cancer Institute, Zhongshan Hospital, Fudan University, Shanghai, China; [5]Department of Neuroscience, Baylor College of Medicine, Houston, United States; [6]Department of Neurosurgery, Beijing Tongren Hospital, Capital Medical University, Beijing, China; [7]Graduate School of Biomedical Sciences, Tufts University, Boston, United States; [8]Clinical Science Division, H. Lee Moffitt Cancer Center & Research Institute, Tampa, United States; [9]Department of Bioinformatics and Computational Biology, The University of Texas MD Anderson Cancer Center, Houston, United States; [10]Neuroscience Program, MD Anderson Cancer Center UTHealth Graduate School of Biomedical Sciences, Houston, United States

*For correspondence:
JHu3@mdanderson.org

†These authors contributed equally to this work

Competing interests: The authors declare that no competing interests exist.

**Abstract** Myelination depends on timely, precise control of oligodendrocyte differentiation and myelinogenesis. Cholesterol is the most abundant component of myelin and essential for myelin membrane assembly in the central nervous system. However, the underlying mechanisms of precise control of cholesterol biosynthesis in oligodendrocytes remain elusive. In the present study, we found that Qki depletion in neural stem cells or oligodendrocyte precursor cells in neonatal mice resulted in impaired cholesterol biosynthesis and defective myelinogenesis without compromising their differentiation into Aspa[+]Gstpi[+] myelinating oligodendrocytes. Mechanistically, Qki-5 functions as a co-activator of Srebp2 to control transcription of the genes involved in cholesterol biosynthesis in oligodendrocytes. Consequently, Qki depletion led to substantially reduced concentration of cholesterol in mouse brain, impairing proper myelin assembly. Our study demonstrated that Qki-Srebp2-controlled cholesterol biosynthesis is indispensable for myelinogenesis and highlights a novel function of Qki as a transcriptional co-activator beyond its canonical function as an RNA-binding protein.

## Introduction

The brain is the most cholesterol-rich organ, accounting for 23% of the total cholesterol in the body even though the brain represents only 2% of the total body weight (*Dietschy and Turley, 2004*). About 70–80% of the brain cholesterol resides in myelin, a compact multilayer membrane structure

that is generated by oligodendrocytes in the central nervous system (CNS), and myelin is critical for rapid saltatory nerve conduction (*Armati and Mathey, 2010*). Myelin is a lipid-rich material, and cholesterol accounts for the highest molar percentage (~52%) among all myelin lipids (*Chrast et al., 2011*). Consistently, the rate of cholesterol biosynthesis in the mouse brain is highest during the first three weeks after birth (*Dietschy and Turley, 2004*; *Quan et al., 2003*), the same period in which the rate of myelinogenesis is highest (*Armati and Mathey, 2010*).

The brain depends extensively on de novo cholesterol biosynthesis, which is mainly carried out by oligodendrocytes and astrocytes, as the blood-brain barrier blocks the uptake of cholesterol from the circulation (*Camargo et al., 2017*; *Saher et al., 2005*). Genetic ablation of squalene synthase (*Fdft1*), an enzyme involved in the early step of cholesterol biosynthesis, in oligodendrocyte lineage cells has resulted in failure of proper myelination (*Saher et al., 2005*). The importance of cholesterol biosynthesis in myelination is also implicated in various neurological disorders accompanied by myelin defects. For example, hereditary diseases such as Smith–Lemli–Opitz syndrome (SLOS), desmosterolosis, and lathosterolosis are caused by mutations of the cholesterol biosynthesis genes encoding 7-dehydrocholesterol reductase (DHCR7), 24-dehydrocholesterol reductase (DHCR24), and sterol-C5-desaturase (SC5D), respectively (*Kanungo et al., 2013*; *Nwokoro et al., 2001*; *Porter and Herman, 2011*). In particular, SLOS patients present with cognitive defects and delayed motor and language development along with hypomyelination (*Porter and Herman, 2011*). Patients with desmosterolosis have defects in the corpus callosum, thinning of white matter, and seizures (*Zerenturk et al., 2013*). One of the major neurodevelopmental disorders, schizophrenia, usually develops in late adolescence or early adulthood when maturation of the brain, including steps such as myelin biogenesis, occurs (*Le Hellard et al., 2010*; *Steen et al., 2017*). White matter abnormalities often occur in schizophrenic patients in association with reduced lipid metabolism (*Steen et al., 2017*). Of note, among the 108 schizophrenia-associated genomic loci, one is on chromosome 22q13.2, which includes *SREBF2* gene that encodes sterol regulatory element-binding protein 2 (SREBP2), the major transcription factor that regulates cholesterol biosynthesis (*Horton et al., 2002*; *Le Hellard et al., 2010*; *Steen et al., 2017*). Antipsychotic drugs are known to increase SREBP2 activity, resulting in upregulated expression of the genes involved in cholesterol biosynthesis (*Fernø et al., 2005*; *Le Hellard et al., 2009*), suggesting a potential role of SREBP2-mediated cholesterol biosynthesis in the pathogenesis of schizophrenia. Besides the neurological diseases accompanied by myelination defects, reduction in cholesterol biosynthesis is also associated with neurodegenerative diseases such as Alzheimer's disease, Huntington's disease, Parkinson's disease, and autism spectrum disorders, for which myelin involvement has been documented but less understood (*Leoni and Caccia, 2014*; *Mohamed et al., 2018*; *Segatto et al., 2019*; *Tsunemi et al., 2012*; *Xiang et al., 2011*). Taken together, cholesterol biosynthesis plays a pivotal role in brain function, particularly myelination, and dysregulated cholesterol metabolism causes various neurological diseases. Yet the underlying mechanisms of precise control of cholesterol biosynthesis in oligodendrocytes during developmental myelination remain elusive.

Mammalian *Quaking* (*Qk*) undergoes alternative splicing to express the RNA-binding proteins Qki-5, Qki-6, and Qki-7 (*Darbelli and Richard, 2016*). Various studies have extensively demonstrated that Qki regulates the RNA processing of the genes encoding myelin basic protein (MBP), myelin-associated glycoprotein (MAG), p27kip1, and neurofascin 155 in oligodendrocytes (*Darbelli et al., 2016*; *Larocque et al., 2005*; *Larocque et al., 2002*; *Li et al., 2000*; *Zhao et al., 2010*). The quaking viable (*qk^v*) mouse is a spontaneous recessive mutant with an approximate 1-Mbp deletion in the upstream of *Qk* locus, leading to diminished expression of Qki in oligodendrocytes (*Ebersole et al., 1996*; *Hardy et al., 1996*). *Qk^v* homozygotes suffer from tremor and early death due to severe hypomyelination in the CNS (*Sidman et al., 1964*). Previous studies showed that *qk^v* mice exhibited reduced myelin lipid content, including cholesterol (*Baumann et al., 1968*; *Singh et al., 1971*), and this phenomenon was thought to be secondary to impaired differentiation and maturation of oligodendrocytes (*Chen et al., 2007*; *Darbelli et al., 2016*; *Larocque et al., 2005*). However, similar numbers of oligodendrocytes in some regions of the CNS in *qk^v* mice and control mice and even hyperplasia in these regions in the former mice were observed (*Doukhanine et al., 2010*; *Hardy et al., 1996*; *Myers et al., 2016*). These contradictory data suggested that Qki might regulate myelination in the CNS through mechanisms besides controlling oligodendrocyte differentiation and that reduced myelin lipid in *qk^v* mice could be a direct consequence of *Qk* loss rather than secondary to impaired oligodendrocyte differentiation. Interestingly, our recent study demonstrated

that Qki is not required for mature oligodendrocyte survival in adult mice, and that Qki-5 forms a complex with peroxisome proliferator-activated receptor beta (PPARβ)-retinoid X receptor alpha (RXRα) to transcriptionally control fatty acid metabolism, which is essential for mature myelin maintenance (*Zhou et al., 2020*). We therefore hypothesized that Qki might transcriptionally regulate lipid metabolism, such as cholesterol biosynthesis, during developmental myelination in young mice.

In the present study, using conditional *Qk*-knockout mice specifically lacking expression of Qki in either neural stem cells (NSCs) or oligodendrocyte precursor cells (OPCs) during the crucial myelin-forming period of postnatal brain development, we identified a critical, previously uncharacterized phenomenon that Qki-depleted NSCs and OPCs can still differentiate into Aspa+/Gstpi+ myelinating oligodendrocytes. However, these cells do not form myelin properly due to impaired cholesterol biosynthesis. We also found that Qki-5 interacted with Srebp2 and activated its transcriptional activity for the genes involved in cholesterol biosynthesis. Deletion of *Qk* almost completely abolished expression of various cholesterol biosynthesis genes in oligodendrocytes and reduced the cholesterol content in the corpus callosum tissues in *Qk*-depleted mice. Our study highlights a novel function of Qki as a co-activator of Srebp2 in developing brain beyond its canonical function as an RNA-binding protein.

## Results

### Qki depletion in mouse NSCs leads to hypomyelination in the CNS

We asked whether the hypomyelination induced by knockout of *Qk* is due to impaired oligodendrocyte differentiation or defective myelinogenesis. Unlike the *Olig2-Cre* and *Pdgfra-CreER*[T2], the *Nestin-CreER*[T2] transgene, in which expression of tamoxifen-inducible Cre is under the control of the *Nestin* promoter, enabled us to investigate whether knockout of *Qk* affects differentiation of the entire oligodendrocyte lineage, including the step from NSCs to OPCs. We crossed mice bearing the *Qk-loxP* allele with mice bearing the *Nestin-CreER*[T2] transgene, and *Qk* was specifically deleted in NSCs by injecting tamoxifen into C57BL/6J *Nestin-CreER*[T2];*Qk*[L/L] pups at postnatal day 7 (P7; in all subsequent experiments using this cohort)—the time point when oligodendrocyte differentiation and myelinogenesis start (*Figure 1A*; *Armati and Mathey, 2010*). About 12 days after tamoxifen injection, *Nestin-CreER*[T2];*Qk*[L/L] mice (hereafter denoted as '*Qk*-Nestin-iCKO mice') began to exhibit visible tremors and ataxia accompanied by a significant reduction in coordinate movement as measured using the rotarod test and a marked growth retardation (*Figure 1B–D*, *Figure 1—figure supplement 1A*, *Video 1*). Neurological deficits in *Qk*-Nestin-iCKO mice progressed quite rapidly, and the mice eventually displayed hunched posture, paralysis, and hyperpnea, with a median survival duration of 13 days after tamoxifen injection (*Figure 1E*). In contrast, *Nestin-CreER*[T2];wild-type (WT), *Nestin-CreER*[T2];*Qk*[L/+], and *Qk*[L/L] littermates, which were also injected with tamoxifen in the same manner, were phenotypically normal. Thus, these littermates were used as control mice in subsequent experiments unless specified otherwise.

Two weeks after tamoxifen injection (in all subsequent experiments unless specified otherwise), *Qk*-Nestin-iCKO mice had severe hypomyelination as evidenced by substantial reduction in the levels of expression of MBP, proteolipid protein (PLP), and MAG (23.7%, 28.2%, and 16.6% of those in control mice, respectively) and a marked increase in the frequency (4.1-fold greater than that in control mice) of Iba1+ microglial infiltration in the corpus callosum tissues (*Figure 1F*). To further determine the effect of Qki on myelin formation, we crossed mice bearing the *mTmG* reporter line (*Muzumdar et al., 2007*) with *Qk*-Nestin-iCKO mice or control mice. The mTmG reporter, in which expression of cell membrane-localized tdTomato (mT) is replaced by cell membrane-localized EGFP (mG) in Cre recombinase-expressing cells, enabled us to trace newly formed myelin (after tamoxifen injection) according to the

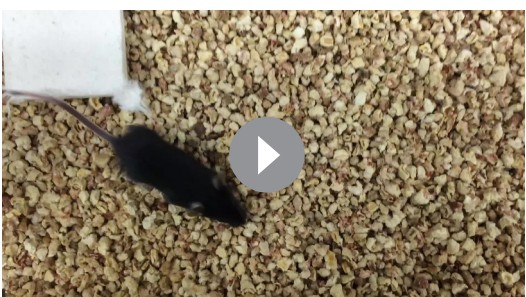

**Video 1.** Defect in motor coordination displaying tremors and ataxia in *Qk*-Nestin-iCKO mice. https://elifesciences.org/articles/60467#video1

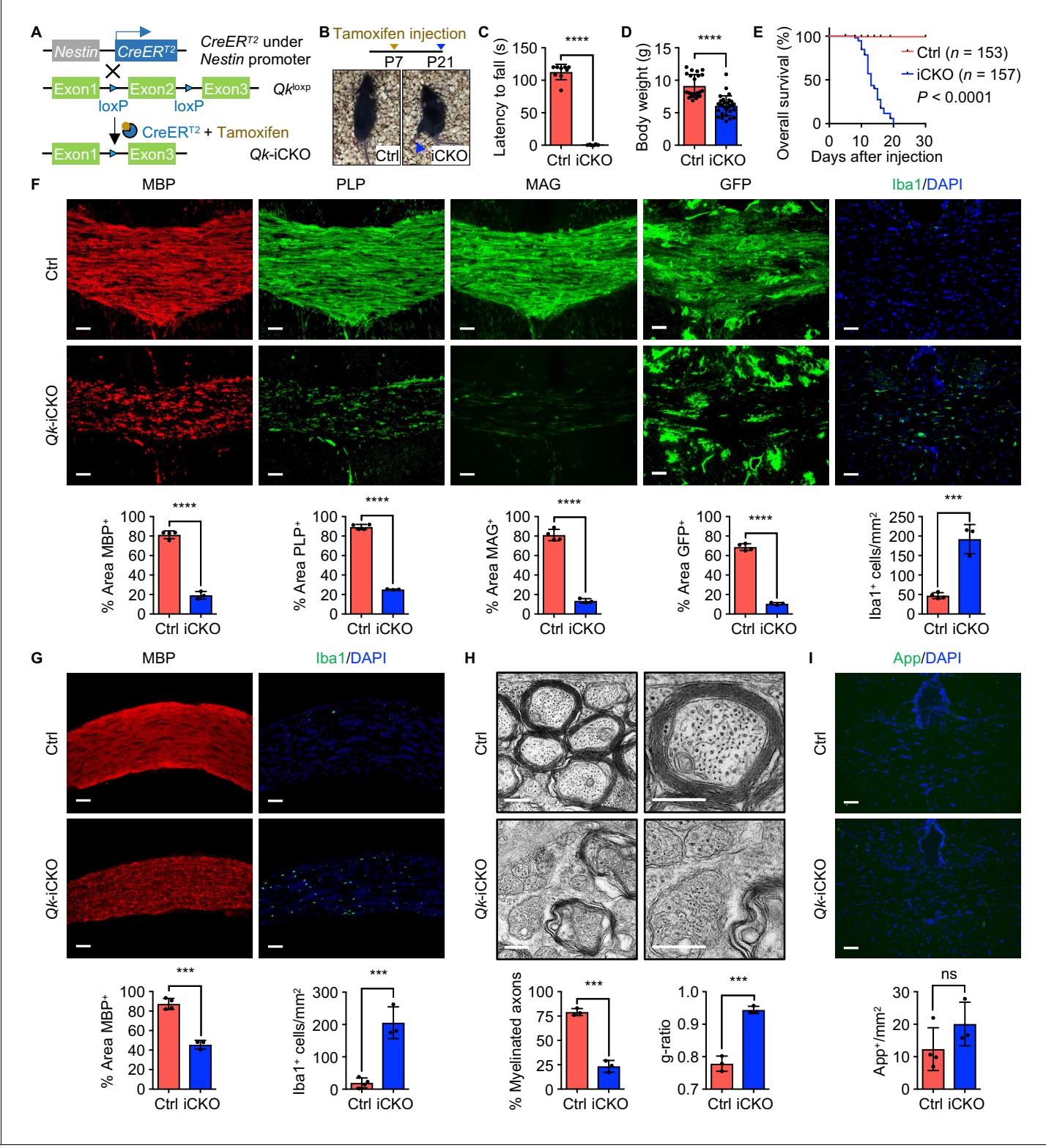

**Figure 1.** Deletion of *Qk* in mouse neural stem cells leads to hypomyelination in the central nervous system. (**A**) Schema of the generation of *Qk*-Nestin-iCKO mice. (**B**) Representative images of severe hind limb paralysis in *Qk*-Nestin-iCKO mice 2 weeks after tamoxifen injection. Ctrl: control. (**C**) Latency of mice falling off the rotarod at a constant speed (5 rpm). *n* = 6 mice in the *Qk*-Nestin-iCKO group; *n* = 9 mice in the control group. (**D**) Body weights of *Qk*-Nestin-iCKO mice (*n* = 29) and control mice (*n* = 22) 12 days after tamoxifen injection. (**E**) Kaplan–Meier curves of and log-rank test results for overall survival in *Qk*-Nestin-iCKO mice (*n* = 157) and control mice (*n* = 153). (**F**) Representative images of and quantification of

*Figure 1 continued on next page*

*Figure 1 continued*

immunofluorescent staining of MBP, PLP, MAG, GFP, and Iba1 in the corpus callosum tissues in *Qk*-Nestin-iCKO mice (*n* = 3) and control mice (*n* = 4) 2 weeks after tamoxifen injection. Scale bars, 50 μm. (**G**) Representative images of and quantification of immunofluorescent staining of MBP and Iba1 in the optic nerves in *Qk*-Nestin-iCKO mice (*n* = 3) and control mice (*n* = 4) 2 weeks after tamoxifen injection. Scale bars, 50 μm. (**H**) Representative electron micrographs of the optic nerves in *Qk*-Nestin-iCKO mice and control mice with quantification of the percentage of myelinated axons and g-ratios 2 weeks after tamoxifen injection (*n* = 3 mice/group). Scale bars, 500 nm. (**I**) Representative images and quantification of immunofluorescent staining of amyloid precursor protein (App) in the corpus callosum tissues in *Qk*-Nestin-iCKO mice (*n* = 3) and control mice (*n* = 4) 2 weeks after tamoxifen injection. Scale bars, 50 μm. Data are shown as mean ± s.d. and were analyzed using Student's *t* test. ***p<0.001; ****p<0.0001; ns: not significant.

The online version of this article includes the following source data and figure supplement(s) for figure 1:

**Source data 1.** Exact p-values for statistical analysis.

**Figure supplement 1.** Deletion of *Qk* in mouse neural stem cells leads to hypomyelination in the central nervous system.

membrane-bound GFP signals. The percentage of the GFP$^+$ area in the corpus callosum tissues in *Qk*-Nestin-iCKO;*mTmG* mice markedly decreased relative to that in control *Nestin-CreER*$^{T2}$;*mTmG* mice (10.6% vs. 68.7%; *Figure 1F*). Similarly, hypomyelination accompanied by microglial infiltration was observed in the optic nerves in *Qk*-Nestin-iCKO mice (*Figure 1G*). In line with these findings, ultrastructural analyses of myelin sheaths in the optic nerves using transmission electron microscopy revealed that only 23.3% of the axons were myelinated in *Qk*-Nestin-iCKO mice, whereas compact myelin formed in 79.0% of the axons in control mice (*Figure 1H*, *Figure 1—figure supplement 1B*). The myelin sheaths wrapping the sparse axons in *Qk*-Nestin-iCKO mice were considerably thinner than those in control mice and failed to form the compact myelin structure (g-ratio, 0.94 vs. 0.78; p<0.0001; *Figure 1H*). We further performed a time-course analysis of MBP expression in the corpus callosum to compare the efficiency of myelin formation between *Qk*-Nestin-iCKO mice and control mice during the critical time of myelin growth (P12–P21). Qki depletion slowed down the myelin formation during P12–P21, a time frame in which myelin is initiated and actively generated, ultimately resulting in failure of proper myelin formation at P21 (*Figure 1—figure supplement 1C*). Notably, we observed that Qki depletion leads to defect in formation of node of Ranvier at P21 in the optic nerve, where rapid and robust myelination occurs during development, using antibodies against paranodal protein (Caspr) and nodal proteins (AnkG and PanNav) (*Figure 1—figure supplement 1D*). As node formation was severely affected at P21, we further examined when the defect is initiated during early myelin development. Previous studies showed that clustering of ion channels at the nodes requires proper myelination (*Rasband and Peles, 2021*; *Rasband et al., 1999*). As myelination is rapidly formed at its peak from P14 in the optic nerve (*Mayoral et al., 2018*), we monitored earlier times (P14 and P17) to ask if myelin defect induced by Qki depletion affects de novo formation of nodes. Defect in node formation was observed as early as P14, which cannot be overcome at P21 (*Figure 1—figure supplement 1E, F*), and this observation is in line with the defect in myelination observed from P12 (*Figure 1—figure supplement 1C*). Specifically, total number of nodes (including both intact and the incomplete nodes) in the optic nerve was decreased upon Qki depletion (*Figure 1—figure supplement 1G*). Importantly, the percentage of intact nodes among the total nodes was significantly reduced with Qki depletion (*Figure 1—figure supplement 1H*). Our observation suggests that the failure of myelin formation upon Qki depletion leads to the failure of node formation during the critical period of myelin development. Despite severely compromised axonal ensheathment and defective structure of node of Ranvier, the axonal diameter and density of axon in *Qk*-Nestin-iCKO mice were comparable with those in control mice (*Figure 1—figure supplement 1I, J*), and no evidence of axonal damage was detected in *Qk*-Nestin-iCKO mice via immunofluorescent staining of amyloid precursor protein (*Figure 1I*). In addition, we further asked if Qki loss impacts on the axon initial segment (AIS) structure. We found that the length of AIS labeled by AnkG at the proximal axon adjacent to the soma (NeuN) was not altered in the cortex region of *Qk*-Nestin-iCKO mice compared to control mice (*Figure 1—figure supplement 1K–M*), further suggesting that the integrity of the axon was not affected upon Qki depletion. Taken together, these data demonstrated that depletion of Qki in mouse NSCs results in severe neurological deficits due to hypomyelination in the CNS accompanied by the defect in node formation.

## Qki loss in NSCs does not impair formation of OPCs or Aspa⁺Gstpi⁺ myelinating oligodendrocytes

Next, we sought to determine whether the hypomyelination observed in *Qk*-Nestin-iCKO mice was caused by a defect in the generation of OPCs from NSCs, impaired maturation from OPCs to oligodendrocytes, or a failure of myelinogenesis. Excluding the first possibility, we found that the number of Pdgfrα⁺ OPCs in the developing cortex tissues in *Qk*-Nestin-iCKO mice was slightly higher than that in control mice (*Figure 2A*), probably due to a compensatory increase in the proliferation of OPCs in response to hypomyelination. Notably, 92.6% of Pdgfrα⁺ OPCs in *Qk*-Nestin-iCKO mice lacked expression of Qki, indicating that Qki does not affect the generation or survival of OPCs. Excluding the second possibility above, immunofluorescent staining of Aspa and Gstpi, two well-established markers of mature oligodendrocytes, revealed that the numbers of Aspa⁺ and Gstpi⁺ mature oligodendrocytes in the corpus callosum tissues in *Qk*-Nestin-iCKO mice were comparable to those in control mice (*Figure 2B, C*). Also, co-immunofluorescent staining of Aspa and Gstpi demonstrated that they represent the same mature oligodendrocyte population (*Figure 2D*). The majority of these Aspa⁺ and Gstpi⁺ oligodendrocytes in *Qk*-Nestin-iCKO mice lacked expression of Qki (*Figure 2B, C*), indicating that OPCs without expression of Qki are still capable of differentiating into Aspa⁺Gstpi⁺ mature oligodendrocytes. Of note, we did not use another commonly used marker, CC-1, in our study because a recent study demonstrated that the CC-1 antibody actually recognizes Qki-7 (*Bin et al., 2016*), raising the concern that CC-1 is not a good marker for labeling mature oligodendrocyte in *Qk*-knockout mice. In fact, the number of CC-1⁺ mature oligodendrocytes in the corpus callosum tissues in *Qk*-Nestin-iCKO mice significantly decreased to 6.7% of that in control mice (*Figure 2—figure supplement 1A*), whereas the number of Aspa⁺Gstpi⁺ oligodendrocytes in *Qk*-Nestin-iCKO mice was similar to that in control mice. The reason for this phenomenon is that the Aspa⁺Gstpi⁺ oligodendrocytes in *Qk*-Nestin-iCKO mice cannot be recognized by CC-1 antibodies due to the absence of Qki-7 in these cells.

Analyses of the previous transcriptomic studies (*Marques et al., 2016*; *Zhang et al., 2014*) revealed that the mRNA level of Aspa in myelinating oligodendrocytes was much higher than that in newly formed oligodendrocytes and OPCs (*Figure 2E, F*). In agreement with this, immunofluorescent staining of Aapa in the corpus callosum tissue in mice at P21 revealed expression of Aspa in myelin sheaths in addition to the cell bodies of oligodendrocytes (*Figure 2G*). Coupled with the observation that Aspa and Gstpi positivities represented the same mature oligodendrocyte population (*Figure 2D*), these data demonstrated that Aspa⁺Gstpi⁺ mature oligodendrocytes represent a subset of myelin-forming oligodendrocytes. Of note, the number of Olig2⁺ (marker of oligodendroglial lineage) cells in the corpus callosum tissues in *Qk*-Nestin-iCKO mice was 50.9% lower than that in control mice (*Figure 2—figure supplement 1B*), suggesting that Qki loss partially blocks OPCs differentiation into Olig2⁺Aspa⁻Gstpi⁻ oligodendroglial lineage cells. Still, numbers of TUNEL positive cells were comparable between *Qk*-Nestin-iCKO and control (*Figure 2—figure supplement 1C*), suggesting that the survival of oligodendroglial lineage cells was not affected upon Qki depletion. Taken together, these data suggested that NSCs without expression of Qki are still capable of generating OPCs and subsequently differentiating into Aspa⁺Gstpi⁺ myelinating oligodendrocytes.

Nestin is expressed in NSCs, which can differentiate into neurons, astrocytes, and oligodendrocytes, so deletion of *Qk* in *Qk*-Nestin-iCKO mice potentially also affects neurons and astrocytes besides oligodendrocytes. Immunofluorescent staining of NeuN (a marker of neurons) revealed comparable numbers of neurons in the brains in *Qk*-Nestin-iCKO mice and control mice (*Figure 2—figure supplement 2A*). Notably, Sox9⁺Gfap⁺GFP⁺ astrocytes only constituted a small population among total Sox9⁺Gfap⁺ astrocytes in both *Qk*-Nestin-iCKO;*mTmG* mice (15.92%) and control *Nestin-CreER^T2*;*mTmG* mice (16.22%) (*Figure 2—figure supplement 2B*), suggesting that the majority of Sox9⁺Gfap⁺ astrocytes are developed prior to P7 and therefore are not targeted by *Nestin-CreER^T2* inducible system with P7 tamoxifen treatment. Collectively, these data suggested that Qki loss in NSCs has minimal or no effect on the neuron and astrocyte populations in the brain, and hypomyelination induced by Qki loss is not secondary to defects in neurons or astrocytes.

## Qki loss leads to defective myelin membrane assembly

The unexpected finding that *Qk*-Nestin-iCKO mice did not have reduced numbers of Aspa⁺Gstpi⁺ mature myelin-forming oligodendrocytes yet exhibited severe myelin defects (*Figure 1*) suggested

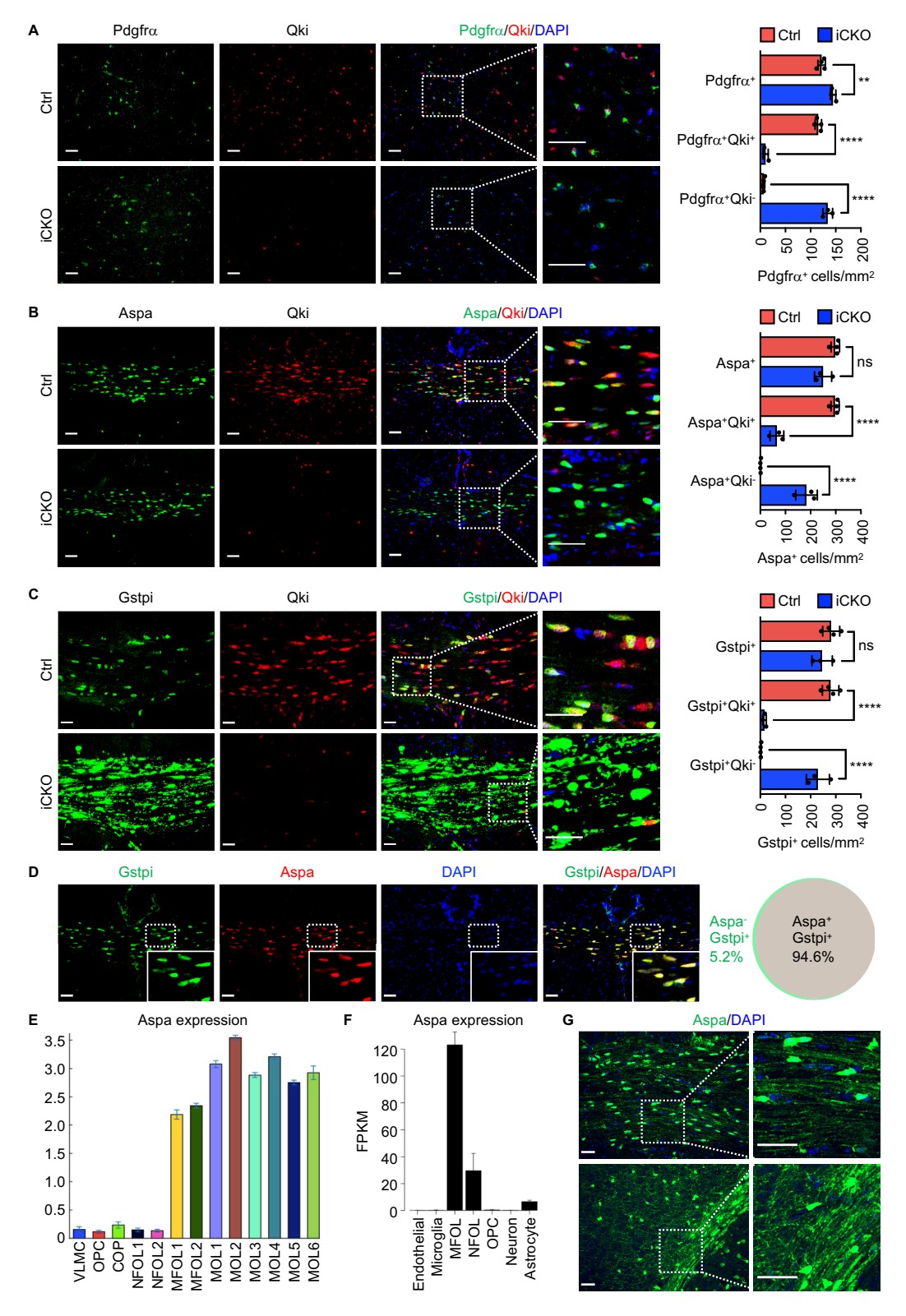

**Figure 2.** Qki loss in neural stem cells does not impair formation of oligodendrocyte precursor cells or Aspa⁺Gstpi⁺myelinating oligodendrocytes. (A–C) Representative images of and quantification of immunofluorescent staining of Pdgfrα-Qki (A), Aspa-Qki (B), and Gstpi-Qki (C) in the corpus callosum tissues in *Qk*-Nestin-iCKO mice (*n* = 3) and control mice (*n* = 4) 2 weeks after tamoxifen injection. Scale bars, 50 μm. (D) Representative images of immunofluorescent staining of Gstpi and Aspa in the corpus callosum tissues in WT mice at 3 weeks of age (*n* = 4 mice/group). Scale bars, 50 μm. The

*Figure 2 continued on next page*

# eLife Research article

Developmental Biology | Neuroscience

*Figure 2 continued*

Venn diagram depicts the overlap of Aspa[+] and Gstpi[+] oligodendrocytes. (**E, F**) RNA-seq expression data for Aspa transcripts from the databases of Gonçalo Castelo-Branco's laboratory (**E**) and Ben A. Barres's and Jiaqian Wu's laboratory (**F**). VLMC: vascular and leptomeningeal cells; COP: differentiation-committed oligodendrocyte precursors; NFOL: newly formed oligodendrocytes; MFOL: myelin-forming oligodendrocytes; MOL: mature oligodendrocytes. (**G**) Representative images of immunofluorescent staining of Aspa in the corpus callosum regions in WT mice at 3 weeks of age. Scale bars, 50 μm. Data are shown as mean ± s.d. and were analyzed using one-way ANOVA with Tukey's multiple comparisons test. **p<0.01; ****p<0.0001; ns: not significant.

The online version of this article includes the following source data and figure supplement(s) for figure 2:

**Source data 1.** Exact p-values for statistical analysis.
**Figure supplement 1.** Qki loss in neural stem cells impairs the differentiation of Olig2[+]AspaGstpi[-] oligodendroglial lineage cells.
**Figure supplement 2.** Deletion of *Qk* in neural stem cells has no effects on neuronal or astrocytic populations.

that Qki loss impairs myelinating ability of Aspa[+]Gstpi[+] oligodendrocytes without affecting the differentiation and survival of these cells. Consistent with previous studies demonstrating that Qki regulates the RNA homeostasis of myelin proteins, including MBP (*Darbelli et al., 2016*; *Larocque et al., 2005*; *Larocque et al., 2002*; *Li et al., 2000*; *Zhao et al., 2010*), we found that expression of MBP in the corpus callosum tissues was greatly reduced upon Qki depletion. In addition, unlike the homogeneous MBP staining in control mice, an uneven and patchy MBP staining pattern was detected in *Qk*-Nestin-iCKO mice (*Figure 3A*), implicating that MBP was not properly assembled in myelin sheaths other than a reduction in its expression. We also found that whereas MBP and PLP co-localized very well in the corpus callosum tissues in control mice, the MBP and PLP co-localization rate markedly decreased in *Qk*-Nestin-iCKO mice (45.7% vs. 88.5%; *Figure 3B*). Similarly, the MBP and MAG co-localization rate in *Qk*-Nestin-iCKO mice was substantially lower than that in control mice (26.9% vs. 75.1%; *Figure 3C*). Previous studies demonstrated that the proper interaction of myelin proteins such as PLP, MBP, and MAG with myelin lipids is required for formation of raft-like domains on myelin membrane, which is essential for myelin membrane assembly (*Aggarwal et al., 2011*; *Ozgen et al., 2016*; *Saher et al., 2005*; *Simons et al., 2000*). To determine whether defective myelin assembly in *Qk*-Nestin-iCKO mice was the result of abnormal myelin lipid component(s), we measured the myelin lipid level in the corpus callosum tissues via staining with FluoroMyelin, a lipophilic dye with high selectivity for myelin lipids (*Monsma and Brown, 2012*). The percentage of the FluoroMyelin[+] area was 88.7% in control mice but only 6.6% in *Qk*-Nestin-iCKO mice (*Figure 3D*). More importantly, although the levels of both PLP and FluoroMyelin in *Qk*-Nestin-iCKO mice were lower than those in control mice, the ratio of FluoroMyelin to PLP in *Qk*-Nestin-iCKO mice was only about one fourth of that in control mice (*Figure 3E*), suggesting that the levels of myelin lipids were more profoundly reduced than were the levels of myelin proteins upon Qki depletion (*Figure 1F*, *Figure 3D*). Of note, the expression of Gstpi was strongly upregulated in the corpus callosum tissues of *Qk*-NSC-iCKO mice (*Figure 2C*). Since Gstpi has been implicated in stress response (*Bartolini and Galli, 2016*), we reasoned that Aspa[+]Gstpi[+] mature oligodendrocytes in *Qk*-Nestin-iCKO mice might cope with imbalanced ratio of myelin lipids to myelin proteins by elevating Gstpi. Taken together, these data suggested that the myelin lipid components are disrupted by Qki loss in oligodendrocytes, which may consequently lead to defective assembly of myelin proteins with myelin lipids and inability to form compact myelin.

## Qki depletion in OPCs leads to defective myelinogenesis without impairing differentiation of Aspa[+] myelinating oligodendrocytes

To further confirm that dysmyelination in *Qk*-Nestin-iCKO mice was oligodendroglial lineage cell-autonomous, mice bearing the *Qk-loxP* allele were bred with mice bearing the *Plp1-CreER*[T2] transgene, in which expression of tamoxifen-inducible Cre is under the control of the *Plp1* promoter (*Figure 4A*). Because the activity of the *Plp1* promoter in the CNS of early neonatal mice is restricted to a subset of OPCs poised to differentiate into myelinating oligodendrocytes (*Guo et al., 2009*), tamoxifen administration in *Plp1-CreER*[T2];*Qk*[L/L] pups at P4 (hereafter denoted as '*Qk*-Plp-iCKO mice') enabled us to delete *Qk* in this subset of OPCs (*Figure 4A*). All *Qk*-Plp-iCKO mice began to experience tremors and ataxia about 11 days after tamoxifen injection, and they gradually displayed reduced coordinate movement, marked growth retardation, hunched posture, and paralysis before dying about 18 days after tamoxifen injection (*Figure 4B–E*, *Figure 4—figure supplement 1A*,

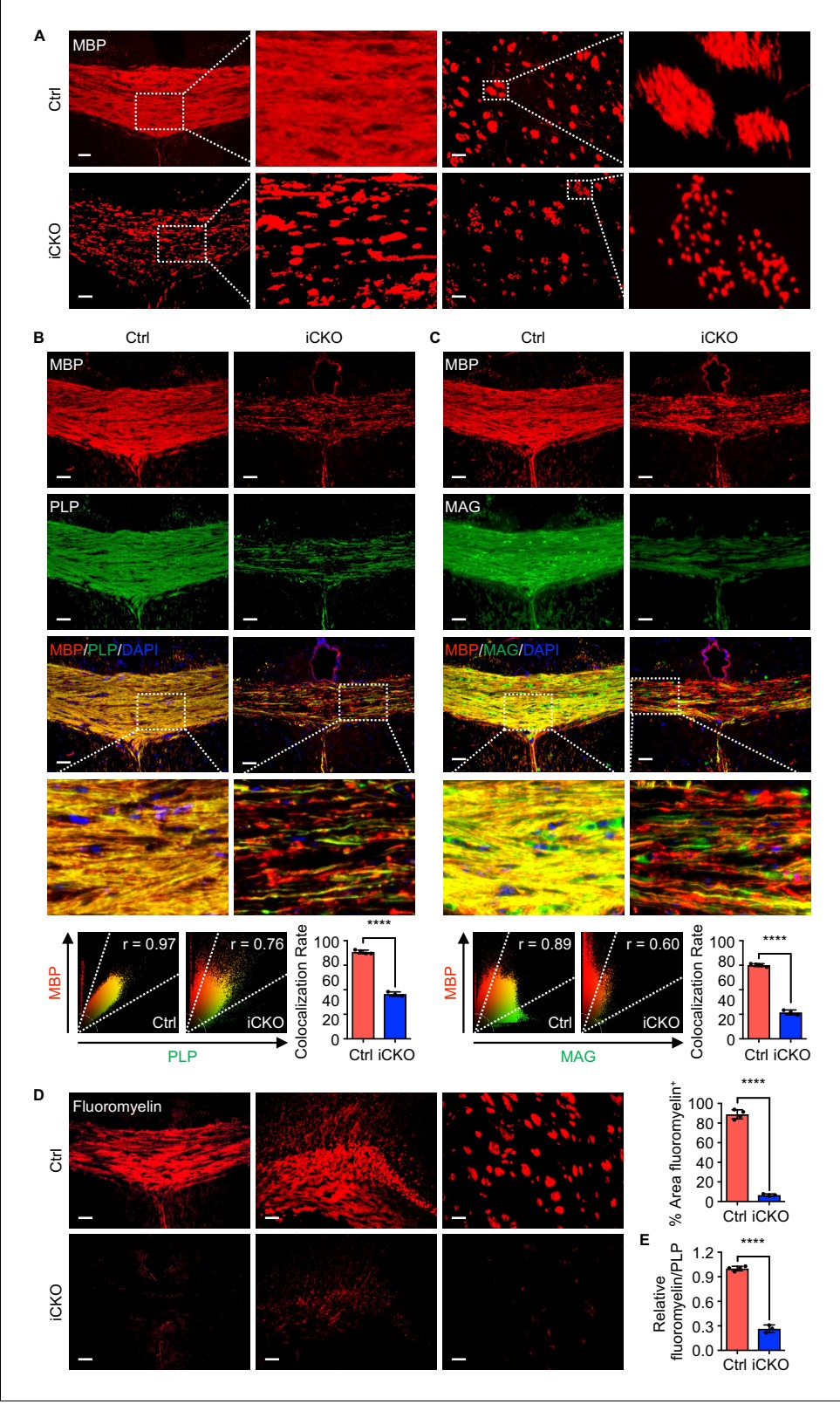

**Figure 3.** Qki loss leads to defective myelin membrane assembly. (**A**) Representative images of immunofluorescent staining of MBP in the corpus callosum tissues in *Qk*-Nestin-iCKO mice and control mice 2 weeks after tamoxifen injection. (**B**, **C**) Representative images of and quantification of the co-localization rates of immunofluorescent staining of MBP-PLP (**B**) and MBP-MAG (**C**) in the corpus callosum tissues in *Qk*-Nestin-iCKO
*Figure 3 continued on next page*

*Figure 3 continued*

mice and control mice 2 weeks after tamoxifen injection. (D) Representative images of and quantification of staining of FluoroMyelin in the corpus callosum tissues in *Qk*-Nestin-iCKO mice and control mice 2 weeks after tamoxifen injection. (E) Quantification of the relative ratio of FluoroMyelin to PLP in the corpus callosum tissues in *Qk*-Nestin-iCKO mice (*n* = 3) and control mice (*n* = 4) 2 weeks after tamoxifen injection. Scale bars, 50 μm. Data are shown as mean ± s.d. and were analyzed using Student's *t* test. The r values in the scatter plots (**B, C**) were calculated using Pearson's correlation. ****p<0.0001.

The online version of this article includes the following source data for figure 3:

**Source data 1.** Exact p-values for statistical analysis.

*Video 2*), whereas the control mice (including *Plp1-CreER*T2;*Qk*+/+, *Plp1-CreER*T2;*Qk*L/+, and *Qk*L/L littermates) did not exhibit neurological symptoms after tamoxifen injection at P4.

Similar to that observed in *Qk*-Nestin-iCKO mice, expression of MBP in the corpus callosum tissues in *Qk*-Plp-iCKO mice 2 weeks after tamoxifen injection (in all subsequent experiments unless specified otherwise) was significantly lower than the robust expression in control mice (*Figure 4F*). In addition, the percentage of the GFP+ area in the corpus callosum tissues in *Qk*-Plp-iCKO;*mTmG* mice markedly decreased relative to that in control *Plp-CreER*T2;*mTmG* mice (5.0% vs. 20.5%; *Figure 4F*), confirming a hypomyelinating phenotype in *Qk*-Plp-iCKO mice. As a secondary response to hypomyelination, three-fold greater accumulation of Iba1+ microglia in the corpus callosum tissues in *Qk*-Plp-iCKO mice than in control mice was observed (*Figure 4F*). Of note, the percentage of the FluoroMyelin+ area was 63.3% in control mice but only 35.2% in *Qk*-Plp-iCKO mice (*Figure 4G*). Ultrastructural analysis of the optic nerves further revealed that when compared with control mice *Qk*-Plp-iCKO mice exhibited a substantially lower percentage of myelinated axons (56.7% vs. 70.4%; *Figure 4H*) and a significantly larger g-ratio (0.86 vs. 0.82; *Figure 4—figure supplement 1B, C*) but a comparable axonal diameter and density of axon (*Figure 4—figure supplement 1D, E*).

The hypomyelinating phenotype in *Qk*-Plp-iCKO mice could be due to compromised oligodendrocyte differentiation or defective myelinogenesis. To determine the effect of Qki on OPC development, we first confirmed that *Plp1-CreER*T2;mTmG cohort labels a subset of OPC population as indicated by the Pdgfrα+GFP+ double-positive cells (*Figure 4—figure supplement 1F*), and Qki loss did not alter the number of Pdgfrα+GFP+ cells (*Figure 4—figure supplement 1F*). Furthermore, no alteration in proliferation was observed upon Qki depletion in the OPC population (Pdgfrα+ cells) and oligodendroglial lineage cells (Olig2+ cells) as indicated by the co-labeling of a proliferating marker, Ki67 (*Figure 4—figure supplement 2A, B*). In addition, comparable numbers of TUNEL positive cells (which are very few) were found between *Qk*-Nestin-iCKO and control (*Figure 2—figure supplement 1C*). These data suggest that the development and survival of OPC population was not altered upon Qki depletion. In addition to the intact OPC survival, the number of Aspa+ mature oligodendrocytes in the corpus callosum tissues in *Qk*-Plp-iCKO mice was comparable to that in control mice, similar to the finding observed in *Qk*-Nestin-iCKO mice (*Figure 2B*), indicating that OPCs with Qki depletion can still differentiate into Aspa+ mature oligodendrocytes (*Figure 4I*). Taken together, these data demonstrated that the hypomyelination induced by Qki depletion in OPCs is attributable to defective myelinogenesis but not OPC survival and oligodendrocyte differentiation.

## Qki regulates transcription of the genes involved in myelin cholesterol biosynthesis

To investigate the underlying mechanisms by which Qki regulates myelinogenesis, transcriptomic profiling (RNA sequencing [RNA-seq]) of the brains in *Qk*-Plp-iCKO mice and control littermates was performed. Overall, expression of 673 and 692 genes in *Qk*-Plp-iCKO mice was significantly lower and higher than those in control mice, respectively (fold change >1.2; p<0.05). Ingenuity pathway analysis (IPA) revealed that the canonical pathways affected most by Qki depletion were cholesterol biosynthesis, mevalonate pathway, zymosterol biosynthesis, and geranylgeranyl diphosphate biosynthesis (*Figure 5A, B*), which are all associated with de novo cholesterol biosynthesis pathway. Similarly, transcriptomic analyses of the brains of *Rosa26-CreER*T2;*Qk*L/L mice and control mice that were treated with tamoxifen at P1 and collected at P7 revealed that lipid metabolism pathways, particularly concentration of sterol and concentration of cholesterol, were among the biological processes

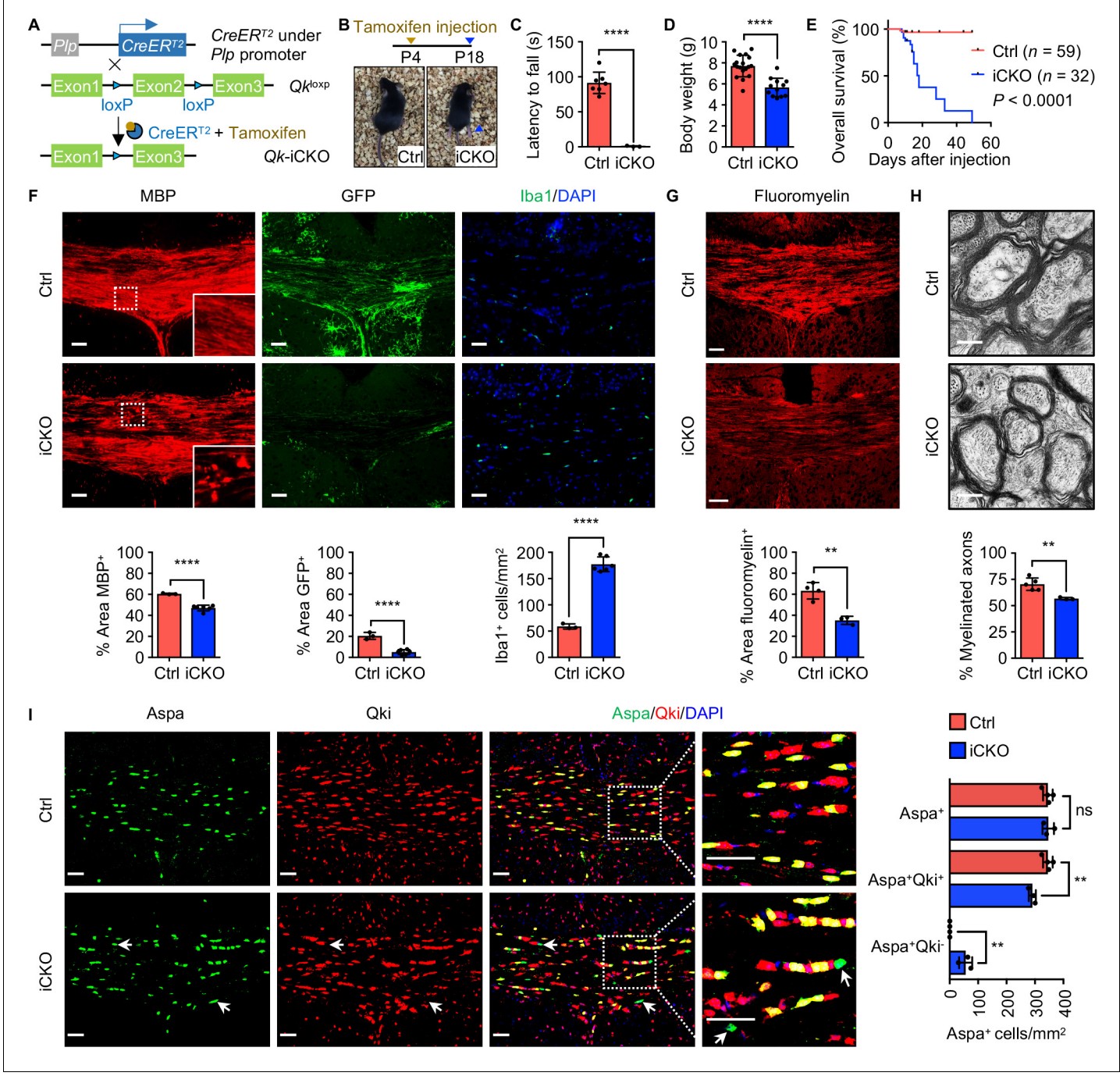

**Figure 4.** *Qk* deletion in oligodendrocyte precursor cells leads to defective myelinogenesis without impairing differentiation of Aspa⁺ myelinating oligodendrocytes. (A) Schema of the generation of *Qk*-Plp-iCKO mice. (B) Representative images of severe hind limb paresis in *Qk*-Plp-iCKO mice 2 weeks after tamoxifen injection. (C) Latency of mice falling off the rotarod at a constant speed (5 rpm). *n* = 3 mice in the *Qk*-Plp-iCKO group; *n* = 7 mice in the control group. (D) Body weights of *Qk*-Plp-iCKO mice (*n* = 12) and control mice (*n* = 18) 2 weeks after tamoxifen injection. (E) Kaplan–Meier curves of and log-rank test results for overall survival in *Qk*-Plp-iCKO mice (*n* = 32) and control mice (*n* = 59). (F) Representative images of and quantification of immunofluorescent staining of MBP, GFP, and Iba1 in the corpus callosum tissues in *Qk*-Plp-iCKO mice (*n* = 6) and control mice (*n* = 3) 2 weeks after tamoxifen injection. Scale bars, 50 μm. (G) Representative images of and quantification of staining of FluoroMyelin in the corpus callosum tissues in *Qk*-Plp-iCKO mice (*n* = 3) and control mice (*n* = 4) 2 weeks after tamoxifen injection. Scale bars, 50 μm. (H) Representative electron micrographs of and quantification of the percentage of myelinated axons in the optic nerves in *Qk*-Plp-iCKO mice (*n* = 3) and control mice (*n* = 5) 2 weeks after tamoxifen injection. Scale bars, 500 nm. (I) Representative images of and quantification of immunofluorescent staining of Aspa and Qki in the corpus callosum tissues in *Qk*-Plp-iCKO mice (*n* = 3) and control mice (*n* = 4) 2 weeks after tamoxifen injection. Scale bars, 50 μm. Data are shown

*Figure 4 continued on next page*

*Figure 4 continued*

as mean ± s.d. and were analyzed using Student's *t* test (**C,D, F–H**) or one-way ANOVA with Tukey's multiple comparisons test (**I**). **p<0.01; ****p<0.0001; ns: not significant.

The online version of this article includes the following source data and figure supplement(s) for figure 4:

**Source data 1.** Exact p-values for statistical analysis.
**Figure supplement 1.** Deletion of *Qk* in mouse oligodendrocyte precursor cells results in hypomyelination in the central nervous system.
**Figure supplement 2.** Deletion of *Qk* does not alter proliferation of oligodendrocyte precursor cells and oligodendroglial lineage cells.

most affected by Qki depletion (*Figure 5—figure supplement 1A*). Consistently, IPA-based upstream regulator analysis revealed that Qki loss led to inactivation of Srebp2 and Srebp cleavage-activating protein (Scap) as well as activation of insulin-induced gene 1 protein (Insig1), which inhibits Srebp2 function by interacting with Scap to retard the Scap/Srebp complex in the endoplasmic reticulum (*Figure 5C*). Hence, transcription of multiple Srebp2 target genes encoding the enzymes involved in cholesterol biosynthesis, including hydroxymethylglutaryl-CoA synthase 1 (*Hmgcs1*), 3-hydroxy-3-methylglutaryl-CoA reductase (*Hmgcr*), farnesyl diphosphate synthase (*Fdps*), and lanosterol synthase (*Lss*), was strongly diminished by Qki depletion (*Figure 5D*). In agreement with the observation in *Qk*-Plp-iCKO mice, quantitative real-time PCR (qPCR) analysis confirmed reduced transcription of these genes involved in cholesterol biosynthesis in *Qk*-Nestin-iCKO mice (*Figure 5E*).

Verifying reduced expression of enzymes involved in cholesterol biosynthesis upon Qki loss, we found that the levels of Hmgcs1 and Fdps proteins, which were measured using immunofluorescent staining in Aspa$^+$Qki$^-$ oligodendrocytes in the corpus callosum tissues in *Qk*-Nestin-iCKO mice, were only 11.2% and 12.7% of those in Aspa$^+$Qki$^+$ oligodendrocytes in control mice, respectively (*Figure 5F, G*). Similarly, the levels of Hmgcs1 and Fdps proteins in Aspa$^+$Qki$^-$ oligodendrocytes in the corpus callosum tissues in *Qk*-Plp-iCKO mice were only 20.2% and 31.2% of those in Aspa$^+$Qki$^+$ oligodendrocytes in control mice, respectively (*Figure 5—figure supplement 1B, C*). Of note, the levels of Hmgcs1 and Fdps in Aspa$^+$Qki$^+$ oligodendrocytes in both *Qk*-Nestin-iCKO mice and *Qk*-Plp-iCKO mice were similar to those in control mice (*Figure 5F, G*, *Figure 5—figure supplement 1B, C*), indicating that expression of Hmgcs1 and Fdps reduced by Qki depletion is oligodendrocyte-autonomous. The lower expression of Hmgcs1, Hmgcs2, Hmgcr, Fdps, and Lss in *Qk*-Nestin-iCKO mice than in control mice at the protein level was further confirmed via immunoblotting (*Figure 5H*). As a consequence of reduced expression of enzymes involved in cholesterol biosynthesis, *Qk*-Nestin-iCKO mice exhibited dramatically lower concentrations of both free cholesterol and cholesteryl ester in the corpus callosum tissues than did control mice (*Figure 5I*). Taken together, these data suggested that Qki regulates transcription of enzymes involved in cholesterol biosynthesis in oligodendrocytes and controls synthesis of this rate-limiting building block of myelinogenesis during development.

## Qki-5 cooperates with Srebp2 to regulate cholesterol biosynthesis

Srebp2 is the major transcription factor that regulates expression of the genes involved in cholesterol biosynthesis and import (*Horton et al., 2002*). Besides lower global expression of the genes involved in cholesterol biosynthesis in Qki-depleted oligodendrocytes than in control oligodendrocytes, we found that Srebp2 and its regulatory partners (Scap and Insig1) were the upstream regulators of these differentially expressed genes (*Figure 5C*). Immunofluorescent staining revealed that expression of Srebp2 in Aspa$^+$ oligodendrocytes was similar in *Qk*-Nestin-iCKO mice and control mice (*Figure 6A*), indicating that transcriptional activity but not expression of Srebp2 may be suppressed by Qki

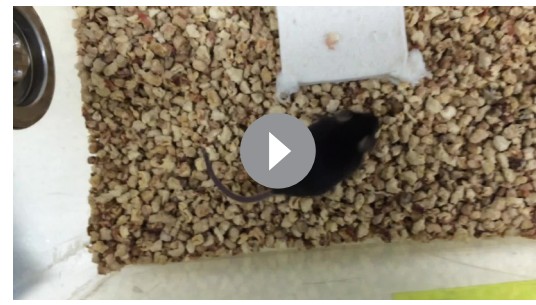

**Video 2.** Defect in motor coordination displaying tremors and ataxia in *Qk*-Plp-iCKO mice.
https://elifesciences.org/articles/60467#video2

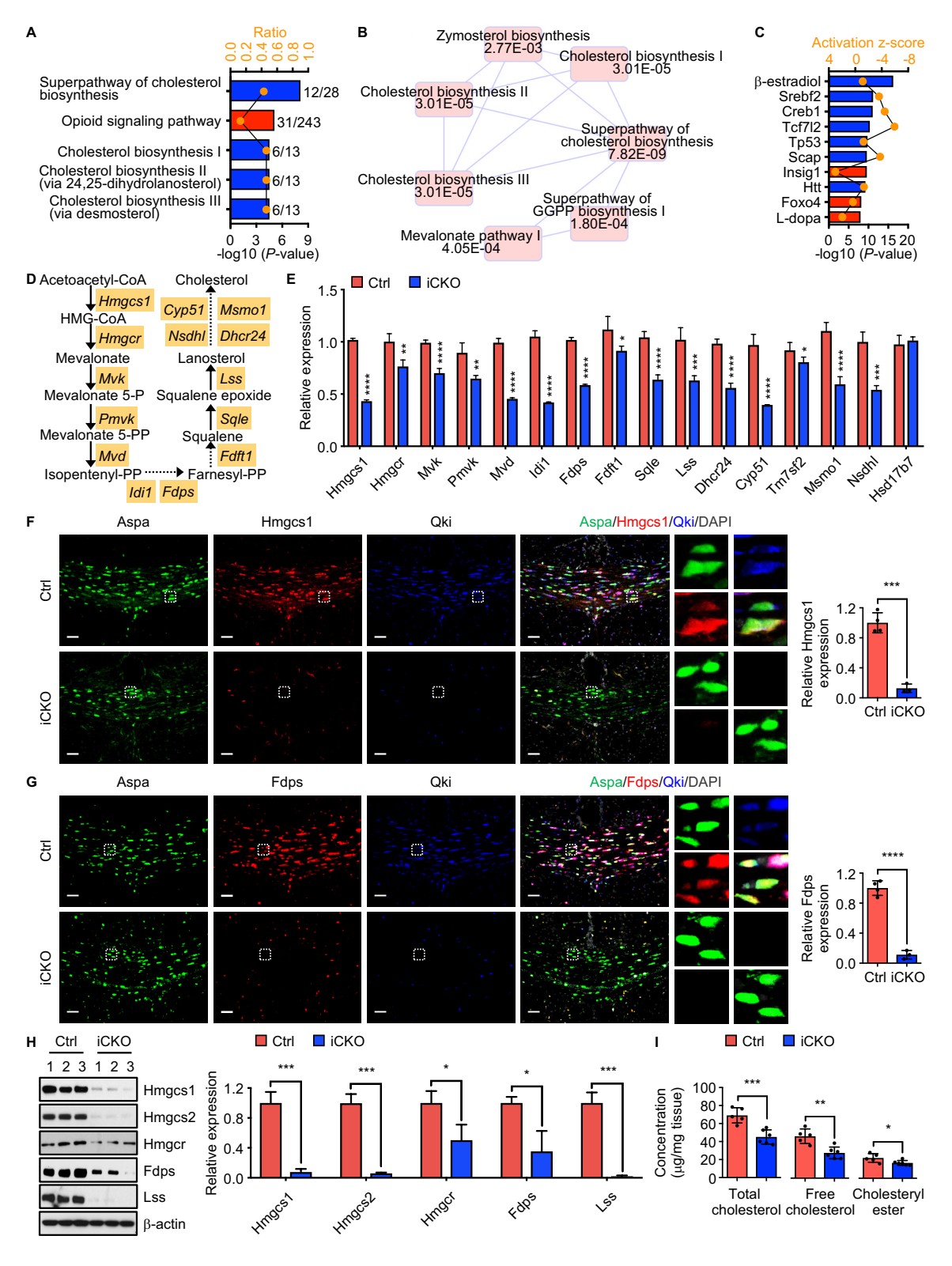

**Figure 5.** Qki regulates transcription of the genes involved in myelin cholesterol biosynthesis. (**A**) Bar graph showing the five canonical pathways most affected by Qki on the basis of differentially expressed genes in *Qk*-Plp-iCKO mice and control mice (*n* = 2 mice/group). Blue and red indicate pathways whose activity decreased or increased, respectively, in *Qk*-Plp-iCKO mice. (**B**) Overlapping canonical pathway networks for the top 20 canonical pathways with a minimum of three common molecules in different pathways. GGPP: geranylgeranyl diphosphate. (**C**) Bar graph showing the

*Figure 5 continued on next page*

Figure 5 continued

10 upstream regulators most enriched in *Qk*-Plp-iCKO mice. (**D**) Schema of the cholesterol biosynthesis pathway. (**E**) Quantification of expression of representative enzymes involved in cholesterol biosynthesis in the corpus callosum tissues in *Qk*-Nestin-iCKO mice and control mice 2 weeks after tamoxifen injection according to real-time qPCR (*n* = 4 mice/group). (**F, G**) Representative images of and quantification of immunofluorescent staining of Hmgcs1 (**F**) and Fdps (**G**) in Aspa⁺Qki⁻ oligodendrocytes in the corpus callosum of *Qk*-Nestin-iCKO mice (*n* = 3) and Aspa⁺Qki⁺ oligodendrocytes in the corpus callosum of control mice (*n* = 4) 2 weeks after tamoxifen injection. Scale bars, 50 μm. (**H**) Immunoblots of and quantification of the levels of expression of the representative enzymes involved in cholesterol biosynthesis in the corpus callosum tissues in *Qk*-Nestin-iCKO mice and control mice 2 weeks after tamoxifen injection (*n* = 3 mice/group). (**I**) Quantification of the cholesterol levels in the corpus callosum tissues in *Qk*-Nestin-iCKO mice (*n* = 6) and control mice (*n* = 5) 2 weeks after tamoxifen injection. Data are shown as mean ± s.d. and were analyzed using Student's *t* test. *p<0.05; **p<0.01; ***p<0.001; ****p<0.0001.

The online version of this article includes the following source data and figure supplement(s) for figure 5:

**Source data 1.** Exact p-values for statistical analysis.
**Figure supplement 1.** Qki regulates transcription of the genes involved in myelin cholesterol biosynthesis.

loss in oligodendrocytes. On the basis of our previous observation that the nuclear-localized Qki isoform, Qki-5 was predominantly localized to chromatin and functions as a co-activator of PPARβ-RXRα complex to transcriptionally regulate fatty acid metabolism in oligodendrocytes in adult mouse brain, which is essential for the maintenance of mature myelin homeostasis (*Zhou et al., 2020*), we hypothesized that Qki-5 regulates transcription of the genes involved in cholesterol biosynthesis by interacting with Srebp2 during myelin development of young mice when cholesterol biosynthesis is highly active. Accordingly, reciprocal co-immunoprecipitation (co-IP) assays performed using oligodendrocytes that differentiated from NSCs revealed a robust interaction between Qki-5 and Srebp2 (*Figure 6B–D*).

To determine whether the molecular interaction between Qki-5 and Srebp2 in oligodendrocytes impacts transcriptional control of the genes involved in cholesterol biosynthesis, we performed chromatin IP (ChIP) combined with high-throughput DNA sequencing (ChIP-seq) using differentiated oligodendrocytes with antibodies against Qki-5 and Srebp2. We identified 17,709 peaks for Qki-5 ChIP-seq and 957 peaks for Srebp2 ChIP-seq. Genomic distribution analyses revealed that 50.11% and 82.27% of Qki-5- and Srebp2-binding events were enriched in the promoter regions, respectively (*Figure 6—figure supplement 1A*). Notably, the promoter/transcriptional start site (TSS)-binding occupancies of Qki-5 and Srebp2 strongly correlated with each other (*Figure 6E, F*). Specifically, 88.73% of total Srebp2-binding events (811 of 914) in the promoter regions overlapped with Qki-5-binding events (*Figure 6G*). Furthermore, ChIP-seq analysis of RNA polymerase II (Pol II), an indicator of transcriptionally active sites, revealed that 99.75% of the overlapping binding events between Qki-5 and Srebp2 (809 of 811) were co-occupied by Pol II (*Figure 6G*), suggesting that the Qki-5 and Srebp2 form a complex that regulates transcription.

Canonical pathway analysis of genes bound by Qki-5, Srebp2, and Pol II (*n* = 809) (*Figure 6G*) revealed that the cholesterol biosynthesis pathway was the most enriched cellular pathway potentially regulated by transcriptional collaboration between Qki-5 and Srebp2 (*Figure 6H*). The promoter regions of the genes involved in cholesterol biosynthesis, such as *Hmgcs1*, *Hmgcr*, *Fdps*, cytochrome P450, family 51 (*Cyp51*), and methylsterol monooxygenase 1 (*Msmo1*), were co-occupied by Qki-5, Srebp2, and Pol II (*Figure 6I*). Consistently, ChIP-qPCR confirmed that Qki-5, Srebp2, and Pol II were highly enriched in the promoter regions of *Hmgcs1* and *Hmgcr* (*Figure 6J*). Taken together, these data suggested that transcriptional cooperation between Qki-5 and Srebp2 regulates cholesterol biosynthesis.

Because Qki-5 binds to a large number of the promoter regions genome-wide, we further sought to identify the specific cellular pathways in oligodendrocytes directly regulated by Qki at transcriptional level during myelin formation. Canonical pathway analysis of 194 overlapping genes among 5885 Qki-5-bound genes in anti-Qki-5 ChIP-seq of freshly isolated mouse oligodendrocytes (*Zhou et al., 2020*) and 673 significantly downregulated genes in *Qk*-Plp-iCKO mice revealed that cholesterol biosynthesis was the most enriched pathway (*Figure 7A, B*). This was also consistent for the overlapping genes among Qki-5-bound genes in differentiated oligodendrocytes and the significantly downregulated genes in *Qk*-Plp-iCKO mice (*Figure 6—figure supplement 1B, C*). This analysis suggested that the Qki-5/Srebp2 complex predominantly regulates expression of the genes involved in cholesterol biosynthesis. Of note, multiple transcription factors such as NF1/CTF, Sox10,

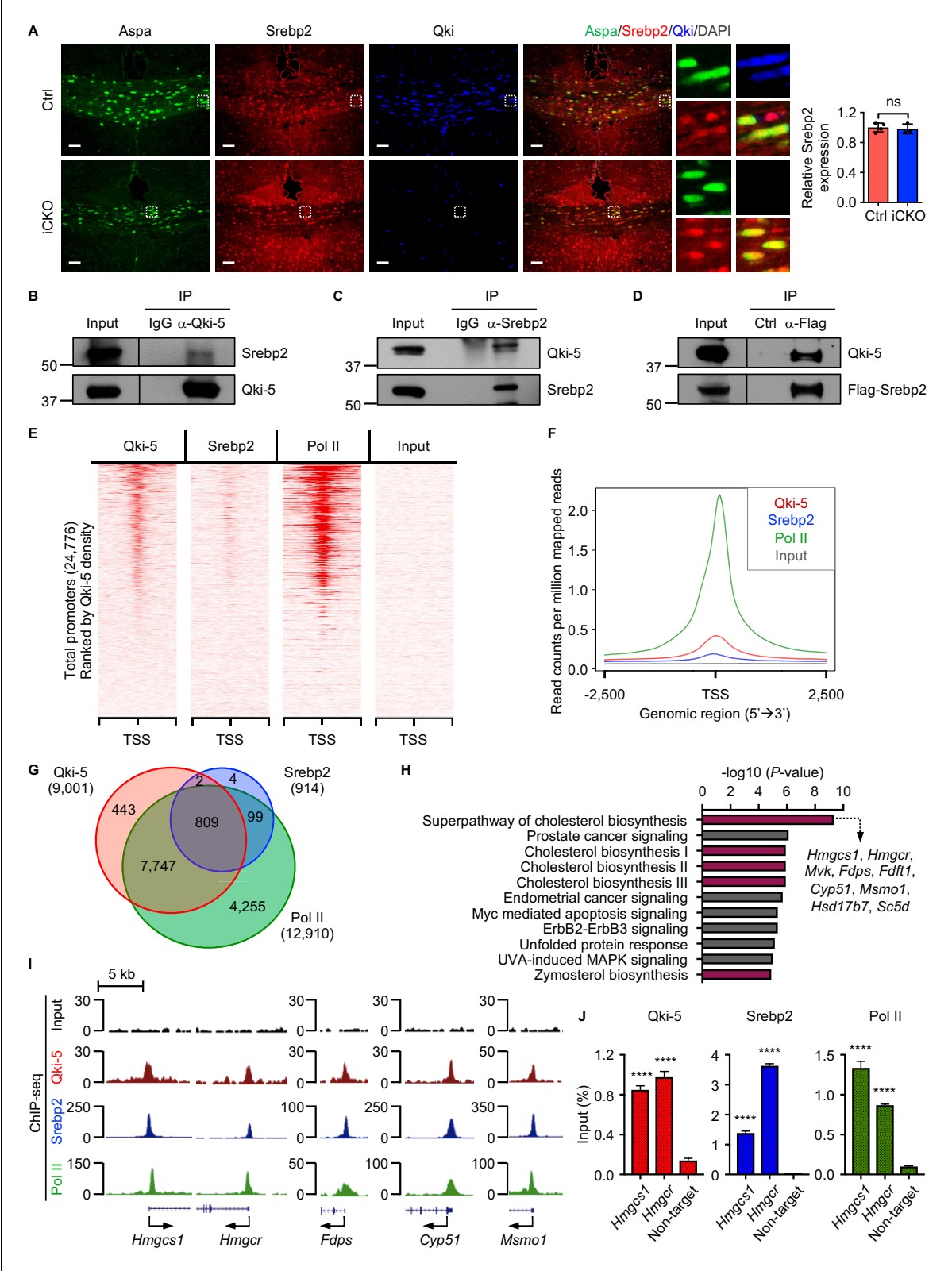

**Figure 6.** Qki-5 interacts with Srebp2 to regulate transcription of the genes involved in cholesterol biosynthesis. (**A**) Representative images of and quantification of immunofluorescent staining of Srebp2 in Aspa⁺Qki⁻ oligodendrocytes in *Qk*-Nestin-iCKO mice (n = 3) and Aspa⁺Qki⁺ oligodendrocytes in control mice (n = 4) 2 weeks after tamoxifen injection. Scale bars, 50 μm. (**B**) Results of co-immunoprecipitation (co-IP) using an anti–Qki-5 antibody with differentiated oligodendrocytes followed by detection of Srebp2 via immunoblotting. (**C**) Results of co-IP using an anti-Srebp2

*Figure 6 continued on next page*

*Figure 6 continued*

antibody with differentiated oligodendrocytes followed by detection of Qki-5 via immunoblotting. (**D**) Results of co-IP using an anti-Flag antibody with differentiated oligodendrocytes having ectopic expression of Flag-Srebp2 followed by detection of Qki-5 via immunoblotting. (**E, F**) ChIP-seq density heat maps (**E**) and average genome-wide occupancies (**F**) of Qki-5, Srebp2, and Pol II in differentiated oligodendrocytes. Regions within 2.5 kb of the transcriptional start site (TSS) are included. All events are rank-ordered from high to low Qki-5 occupancy. (**G**) Venn diagram of the overlap of Qki-5-, Srebp2-, and Pol II-binding events in the promoter regions in differentiated oligodendrocytes. Promoters are defined as TSS ±2 kb. (**H**) Canonical pathway analysis of Qki-5-, Srebp2-, and Pol II-co-occupied genes in differentiated oligodendrocytes. Cellular pathways involved in cholesterol biosynthesis are labeled in dark pink. (**I**) Representative ChIP-seq binding events of Qki-5, Srebp2, and Pol II in the promoter regions of the genes involved in cholesterol biosynthesis. y-axis: normalized reads. (**J**) ChIP-qPCR results showing the recruitment of Qki-5, Srebp2, and Pol II to the promoter regions of *Hmgcs1* and *Hmgcr* in differentiated oligodendrocytes. Data are shown as mean ± s.d. and were analyzed using Student's *t* test. ****p<0.0001; ns: not significant.

The online version of this article includes the following source data and figure supplement(s) for figure 6:

**Source data 1.** Exact p-values for statistical analysis.
**Figure supplement 1.** Qki-5 cooperates with Srebp2 to regulate transcription of the genes involved in cholesterol biosynthesis.

C/EBPα, Rfx1, and Nrf1 were enriched as potential binding partners for Qki-5 based on the motif analysis using Qki-5 ChIP-seq data (***Figure 6—figure supplement 1D***). These transcription factors potentially play important roles in regulating their target genes by cooperating with Qki-5. However, most of the target genes under control of these transcription factors are not altered at the gene expression level in oligodendrocytes. Still, this result provides an important future perspective in studying the function of Qki in transcriptional regulation in other cell types/other tissues.

To determine how Qki-5 transcriptionally enhances Srebp2-mediated cholesterol biosynthesis, we performed ChIP-seq with WT and Qki-depleted differentiated oligodendrocytes using antibodies against Qki-5, Srebp2, and Pol II. Because Srebp2-bound genes were mostly involved in cholesterol biosynthesis (***Figure 6—figure supplement 1E***), to further examine the molecular alteration upon Qki depletion, we focused on the gene clusters bound by Srebp2, which were strongly co-occupied by Qki-5, Srebp2, and Pol II in their promoter regions (***Figure 7C***). Notably, occupancies of Srebp2 (725 of 914) and Pol II (894 of 914) in the promoter regions were strikingly reduced upon Qki depletion (***Figure 7D, E***). Particularly, the promoter occupancy of Srebp2 on all 17 target genes enriched in cholesterol biosynthesis according to IPA analysis (***Figure 6—figure supplement 1E***) such as *Msmo1*, *Hmgcs1*, *Cyp51*, isopentenyl-diphosphate delta isomerase 1 (*Idi1*), squalene epoxidase (*Sqle*), and *Hmgcr* was globally reduced upon Qki depletion (***Figure 6—figure supplement 1F***, ***Figure 7F, G***). ChIP-qPCR further confirmed reduced recruitment of Srebp2 to the promoter regions of *Hmgcs1* and *Hmgcr* upon Qki depletion (***Figure 7H***). Taken together, these data suggested that Qki-5 transcriptionally activates Srebp2-mediated cholesterol biosynthesis in oligodendrocytes, which is essential for proper myelin formation during brain development.

## Discussion

Timely onset of oligodendrocyte myelination is essential for brain development (***Armati and Mathey, 2010***). Cholesterol is an important functional component of myelin formation, and deficiency in cholesterol biosynthesis is associated with various hypomyelinating diseases reported in human genetic studies (***Nwokoro et al., 2001***; ***Porter and Herman, 2011***). Brain cholesterol production mainly depends on de novo synthesis to fulfill the high demand for cholesterol due to the restriction of cholesterol entry into the brain by the blood-brain barrier. However, how cholesterol biosynthesis is regulated in oligodendrocytes, the primary myelinating cells in the CNS, is not clear. The present study revealed that Qki functions as a novel transcriptional co-activator of Srebp2 in oligodendrocytes to ensure supply of cholesterol for proper developmental myelination in a timely manner (***Figure 8***).

Previous studies showed that cholesterol biosynthesis is highly active during the early stage of brain development (***Dietschy and Turley, 2004***). However, the cell types primarily responsible for cholesterol biosynthesis in myelin formation were not clear. In the present study, we observed that Aspa and Gstpi were co-expressed with Srebp2, Fdps, and Hmgcs1 in oligodendrocytes, suggesting that Aspa[+]Gstpi[+] cells represent a subset of myelinating oligodendrocytes with highly active cholesterol biosynthesis (***Figure 5F, G***). Aspa has been shown to be more abundantly expressed in

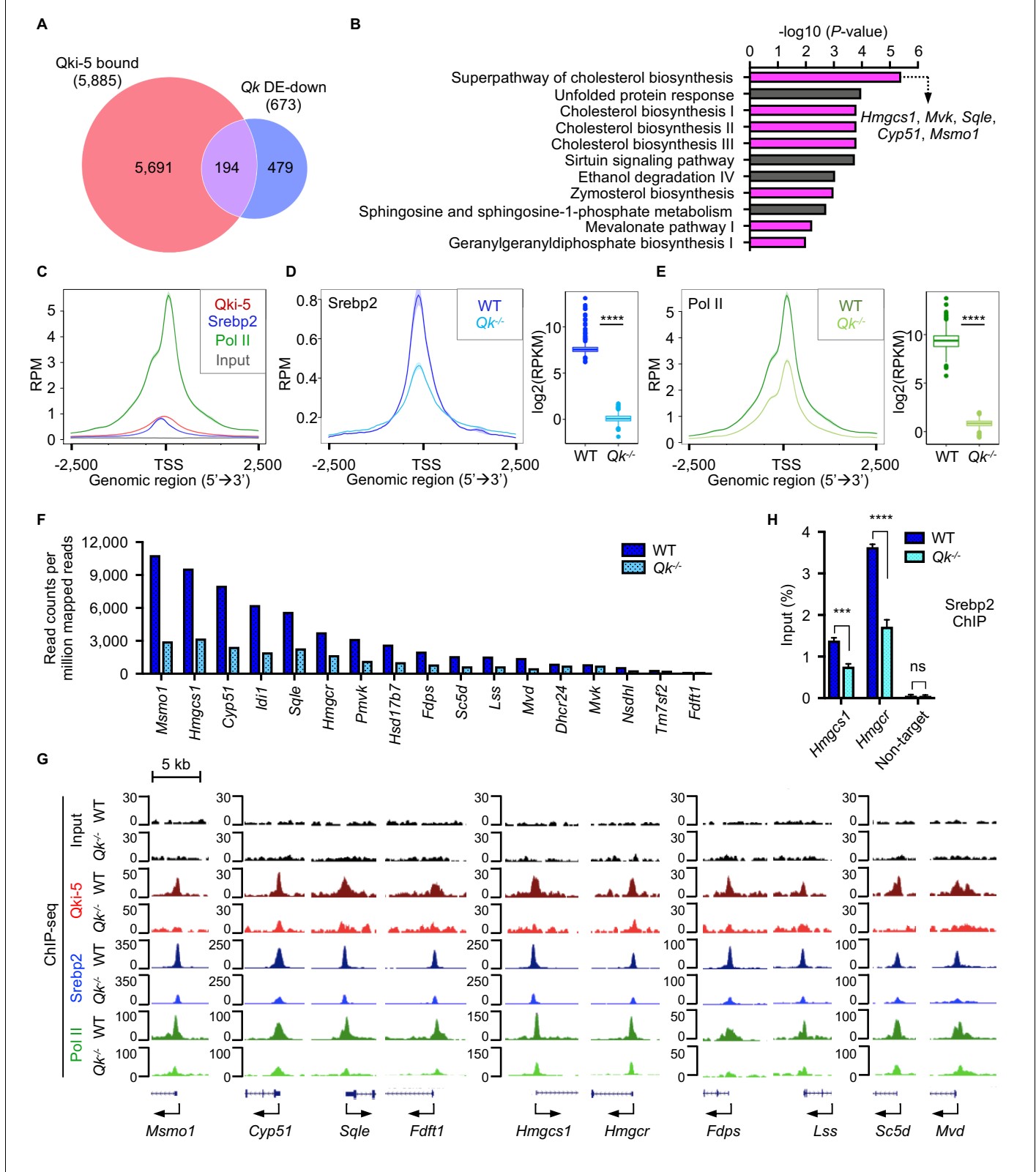

**Figure 7.** Qki transcriptionally enhances Srebp2-mediated cholesterol biosynthesis. (A) Venn diagram of the overlap of Qki-5-bound genes in Qki-5 ChIP-seq from freshly isolated mouse oligodendrocytes and the genes with markedly lower expression in *Qk*-Plp-iCKO mice than in control mice. DE: differentially expressed. (B) Canonical pathway analysis of the 194 overlapping genes shown in (A). Cellular pathways involved in cholesterol biosynthesis are labeled in magenta. (C) Average occupancies of Qki-5, Srebp2, and Pol II in the gene clusters bound by Srebp2 (n = 914) in

*Figure 7 continued on next page*

*Figure 7 continued*

differentiated oligodendrocytes. Regions within 2.5 kb of the transcriptional start site (TSS) are included. (D, E) Average occupancies of Srebp2 (D) and Pol II (E) in the gene clusters bound by Srebp2 in WT and $Qk^{-/-}$ differentiated oligodendrocytes (left) and comparison of ChIP-seq (right). Regions within 2.5 kb of the TSS are included. RPM: reads counts per million mapped reads; RPKM: reads counts per kilobase per million mapped reads. (F) Bar graphs of the RPM of the Srebp2 ChIP-seq peaks within ± 0.5 kb from the TSS for 17 well-characterized Srebp2 target genes involved in cholesterol biosynthesis in WT and $Qk^{-/-}$ differentiated oligodendrocytes. (G) Representative ChIP-seq binding events of Qki-5, Srebp2, and Pol II in the promoter regions of the genes involved in cholesterol biosynthesis in WT and $Qk^{-/-}$ differentiated oligodendrocytes. y-axis: normalized reads. (H) ChIP-qPCR results showing the recruitment of Srebp2 to the promoter regions of *Hmgcs1* and *Hmgcr* in WT and $Qk^{-/-}$ differentiated oligodendrocytes. Data are shown as mean ± s.d. and were analyzed using Student's *t* test. ***p<0.001; ****p<0.0001; ns: not significant.

The online version of this article includes the following source data for figure 7:

**Source data 1.** Exact p-values for statistical analysis.

myelinating oligodendrocytes than in premyelinating oligodendrocytes (*Marques et al., 2016*; *Zhang et al., 2014*). Additionally, Canavan disease caused by *Aspa* deficiency is accompanied by myelin deficiency with reduced level of myelin lipids, including cholesterol, as Aspa enzymatically produces acetate as a source of acetyl-CoA, a precursor for synthesis of cholesterol (*Madhavarao et al., 2005*). These lines of evidence, in combination with our observations, suggest that Aspa⁺Gstpi⁺ cells represent a subset of myelinating oligodendrocytes. In addition to oligodendrocytes, previous studies reported that cholesterol generated by astrocytes also contributes to myelination (*Camargo et al., 2017*). However, the uptake of cholesterol from astrocytes was not sufficient to compensate for the reduced cholesterol synthesis in oligodendrocytes with deletion of

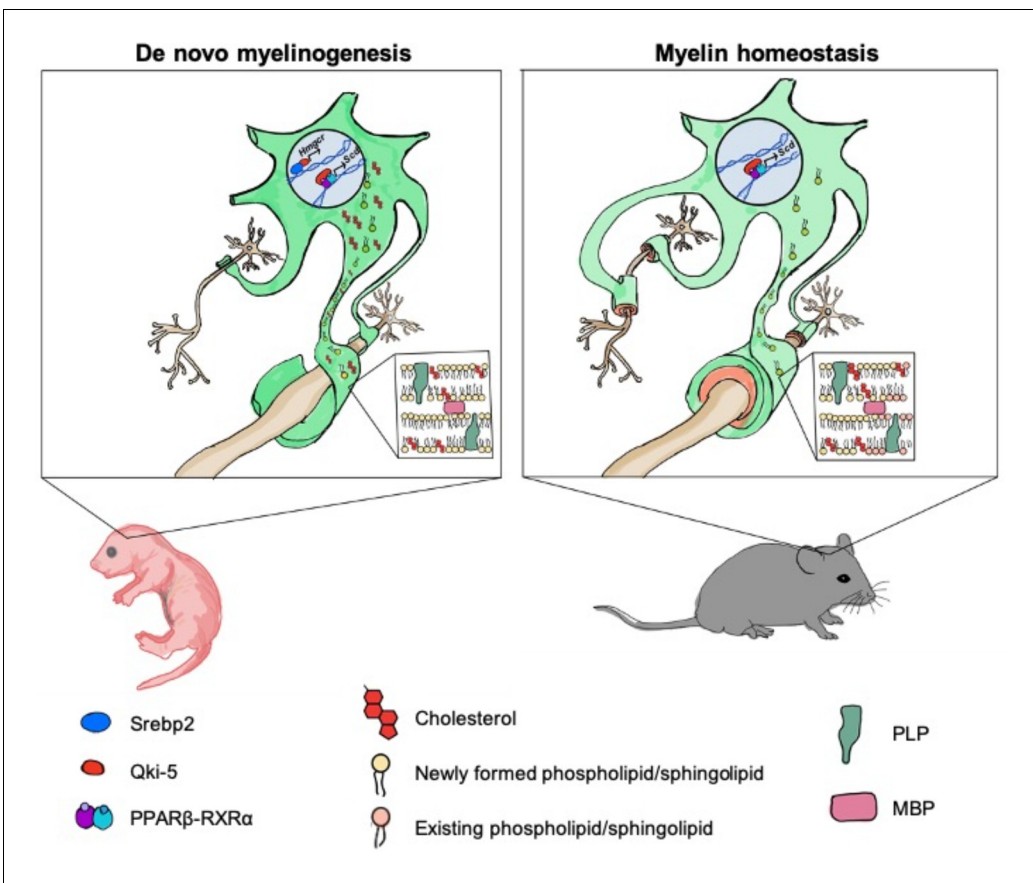

**Figure 8.** Model of Qki's roles in regulating cholesterol biosynthesis and fatty acid metabolism during central nervous system myelination and myelin maintenance. Qki regulates cholesterol biosynthesis in a Srebp2-dependent manner during de novo myelinogenesis but not during myelin maintenance. In contrast, Qki regulates fatty acid metabolism during both de novo myelinogenesis and mature myelin maintenance.

*Scap* (*Camargo et al., 2017*), *Fdft1* (*Saher et al., 2005*), or *Qk* (current study), indicating that oligodendrocytes are the major cell types producing cholesterol for myelination during early development.

Previous studies showed that $qk^v$ mice had reduced myelin lipid content, including cholesterol (*Baumann et al., 1968*; *Singh et al., 1971*). This phenomenon was previously thought to be secondary to loss of mature oligodendrocytes in $qk^v$ mice during development. However, in the present study, we uncovered a previously uncharacterized function of Qki in controlling transcription of the genes involved in cholesterol biosynthesis without affecting the differentiation of Aspa$^+$Gstpi$^+$ myelinating oligodendrocytes. Further studies are needed to elucidate how Aspa$^+$Gstpi$^+$ oligodendrocytes specifically regulate cholesterol biosynthesis and how other cell types, including astrocytes, contribute to oligodendroglial myelination.

During the characterization of oligodendroglial lineage cell populations, we observed that the differentiation of Aspa$^+$Gstpi$^+$ myelinating oligodendrocytes was not impaired upon Qki depletion, but their cholesterol biosynthesis was severely defective. Previous studies showed that all three isoforms of Qki are essential for the differentiation and maturation of oligodendrocytes (*Chen et al., 2007*; *Darbelli et al., 2016*; *Larocque et al., 2005*). Specifically, the number of Olig2$^+$ oligodendroglial lineage cells in $QKI^{FL/FL}$;$^{Olig2-Cre}$ mice was about 50% lower than that in control mice (*Darbelli et al., 2016*). Therefore, we also checked whether oligodendroglial lineage cell populations other than Aspa$^+$Gstpi$^+$ myelinating oligodendrocytes were affected by Qki loss in our *Qk*-Nestin-iCKO mice. We found that the number of Olig2$^+$ cells was reduced by 50.9% in *Qk*-Nestin-iCKO mice compared to that in control mice (*Figure 2—figure supplement 1B*), suggesting that Qki loss affects OPCs differentiation into Olig2$^+$Aspa$^-$Gstpi$^-$ oligodendroglial lineage cells, whose function is unclear. Collectively, we found that Qki plays variable roles in the differentiation of different subpopulations of oligodendrocyte lineage cells, leading to an intriguing question what determines the specific roles of Qki during oligodendrocyte differentiation and myelinogenesis, which needs to be further investigated.

Mammalian *Quaking* (*Qk*) undergoes alternative splicing to express the RNA-binding proteins Qki-5, Qki-6, and Qki-7 (*Darbelli and Richard, 2016*). In the current study, we showed that Qki-5 is required for transcriptional activation of Srebp2-mediated cholesterol biosynthesis in oligodendrocytes. Notably, we observed that expression of myelin proteins such as MBP, PLP, and MAG was greatly reduced upon Qki depletion (*Figure 1F*). Although we did not find these proteins to be direct transcriptional targets of Qki-5 and Srebp2, the stability and nuclear export of MBP mRNA may be regulated by Qki-6 and Qki-7 as shown in $qk^v$ mice, where expression of Qki-6 and Qki-7 is predominantly depleted (*Larocque et al., 2002*; *Li et al., 2000*). Another nonmutually exclusive explanation of decreased expression of MBP is that the reduction in cholesterol level induced by Qki depletion potentially leads to destabilization of myelin because protein:lipid ratio is crucial for proper integration of proteins and lipids in myelin membrane (*Aggarwal et al., 2011*; *Ozgen et al., 2016*; *Saher et al., 2005*; *Simons et al., 2000*), which is shown by the defect in co-localization of different myelin proteins (*Figure 3B*). Although we showed that Qki-5 transcriptionally regulates cholesterol biosynthesis, it is still unclear if Qki-6 and Qki-7 also play a role in the enhancement of cholesterol biosynthesis pathway by regulating mRNA of cholesterol biosynthesis-related genes as we observed more significant decrease of these genes at the protein level compared to the mRNA level (*Figure 5E–H*). Therefore, further investigation is needed to determine the specific roles of different Qki isoforms during oligodendrocyte differentiation and myelinogenesis in early brain developmental stages.

Previous studies showed that cholesterol biosynthesis in oligodendrocytes and Schwann cells could be regulated by transcriptional regulators such as Tcf7l2, Chd8, and Maf (*Kim et al., 2018*; *Zhao et al., 2016*; *Zhao et al., 2018*). However, how Srebp2, the master transcription factor in cholesterol biosynthesis, is regulated in oligodendrocytes is unclear. In the present study, we demonstrated that Qki-5 functions as a co-activator of Srebp2 to enhance transcription of the genes involved in cholesterol biosynthesis in myelinating oligodendrocytes and that depletion of Qki leads to reduced promoter occupancy of Srebp2 and Pol II and decreased transcription of the genes involved in cholesterol biosynthesis (*Figure 7*). Further studies are needed to determine how Qki enhances transcriptional activity of Srebp2.

In this study, we found that Qki-5 functions as a co-activator of Srebp2 to regulate transcription of the genes involved in cholesterol biosynthesis, which is essential for proper myelinogenesis during

development. Although cholesterol biosynthesis was the major downstream of Qki-5, we noticed that the molecular functions related to fatty acid metabolism, concentration of fatty acid, and synthesis of fatty acid were also markedly inhibited after Qki depletion in neonatal mice (*Figure 5—figure supplement 1A*). We assume that both disturbed cholesterol biosynthesis and fatty acid metabolism upon Qki depletion contributed to the hypomyelination and neurological deficits of *Qk*-iCKO mice during development (*Figure 8*). Interestingly, our recent study demonstrated that Qki-5 forms a complex with PPARβ-RXRα and coactivates transcription of the genes for fatty acid metabolism, which is essential for the maintenance of mature myelin homeostasis (*Zhou et al., 2020*; *Figure 8*). However, continuous biosynthesis of cholesterol may not be necessary to maintain the homeostasis of mature myelin, probably due to the very slow turnover of myelin cholesterol in adult rodents (*Ando et al., 2003*; *Smith, 1968*; *Figure 8*). It remains unclear what is the determinant factor of Qki-5 interaction and co-activation of Srebp2 or PPARβ-RXRα on the chromatin under different contexts including the brains from mice of different ages, and whether Qki-5 can interact with other transcription factors. Taken together, the universal occupancy of Qki-5 on chromatin greatly expands our understanding of Qki itself, as well as the biology of transcriptional control by an RNA-binding protein.

Notably, we observed that Qki was depleted in nearly all Pdgfrα$^+$ OPCs (92.6%) in *Qk*-Nestin-iCKO mice (*Figure 2A*), which led to the question of how most of the OPCs were targeted under the control of *Nestin*-Cre after birth. Cortically derived OPCs are considered to make up about 80% of all OPCs in the cortex, which are called late-born OPCs because they arise as a third wave of OPCs initiating at birth and reaching maximum in number 2–4 weeks after birth in mice (*van Tilborg et al., 2018*). This population of OPCs is known to replace the OPCs originating at embryonic stages in the forebrain (*Bergles and Richardson, 2015*), suggesting an active generation of OPCs from NSCs during the first month after birth, potentially leading to highly efficient Qki depletion in NSC-derived Pdgfrα$^+$ OPCs in our study. On the basis of these findings, it is not difficult to understand that although Qki was efficiently depleted in OPCs of *Qk*-Plp-iCKO mice at P4 and P5, an active generation of WT OPCs from WT NSCs diluted the OPC pool, leading to Qki depletion in only a subset of OPCs in *Qk*-Plp-iCKO mice at P18 (*Figure 4I*). Thus, although both *Qk*-Nestin-iCKO mice and *Qk*-Plp-iCKO mice showed dysmyelinating phenotype, *Qk*-Nestin-iCKO mice exhibited a more severe phenotype than did *Qk*-Plp-iCKO mice.

In this study, the median survival duration in *Qk*-Nestin-iCKO mice was 13 days after tamoxifen injection due to severe dysmyelination, yet in our previous study (*Shingu et al., 2017*), *Qk*-Nestin-iCKO mice could survive through adulthood. We believe that the difference in the survival times between these two cohorts of *Qk*-Nestin-iCKO was due to the background of the mice. In the present study, the mice were on a pure C57BL/6J background, whereas the mice in the previous study were on a mixed C57BL/6J and FVB background. OPCs have been shown to repopulate as a third wave of cortically arising OPCs from NSCs initiating at birth and reaching the maximum number around 2–4 weeks after birth (*van Tilborg et al., 2018*). However, the exact time point of this wave of OPC generation can be greatly influenced by the mouse background (*van Tilborg et al., 2018*), resulting in highly variable *Qk* deletion in OPCs with P7 injection of tam in *Qk*-Nestin-iCKO cohorts of different backgrounds.

Effort has been made to treat hypomyelinating disorders such as SLOS, Pelizaeus–Merzbacher disease, and Charcot–Marie–Tooth disease type 1A with dietary supplementation of lipids, including cholesterol (*Fledrich et al., 2018*; *Saher et al., 2012*; *Tierney et al., 2010*). However, challenges still remain, as cholesterol rarely crosses the blood-brain barrier, and long-term benefits of cholesterol supplementation have not been clearly examined (*Tierney et al., 2010*). Whether dietary supplementation of cholesterol can rescue hypomyelination in *Qk*-iCKO mice would be interesting to know. Of note, we previously demonstrated that Qki plays an important role in intracellular vesicle trafficking in NSCs (*Shingu et al., 2017*). If the same function of Qki exists in oligodendrocytes, cholesterol transport and uptake would be impaired due to defective intracellular vesicle trafficking in Qki-depleted oligodendrocytes, which may lead to inefficiencies in high-cholesterol diet-based therapy. The present findings will potentially provide understanding of brain-specific cholesterol biosynthesis and shed light on mechanism-based therapeutics for enhancing oligodendrocyte myelination. As *QKI* is one of the genes most associated with various neurological diseases (*Darbelli and Richard, 2016*), understanding the mechanisms underpinning Qki/Srebp2-mediated cholesterol biosynthesis

and myelination will help identify tissue-specific therapeutic opportunities for neurological diseases associated with myelin defects (*Nwokoro et al., 2001*; *Porter and Herman, 2011*).

# Materials and methods

## Key resources table

| Reagent type (species) or resource | Designation | Source or reference | Identifiers | Additional information |
|---|---|---|---|---|
| Antibody | Mouse monoclonal anti-MBP | BioLegend | Cat# SMI-94R | |
| Antibody | Mouse monoclonal anti–β-Amyloid | BioLegend | Cat# SIG-39220 | |
| Antibody | Rabbit polyclonal anti-PLP | Abcam | Cat# ab105784 | |
| Antibody | Goat polyclonal anti-Iba1 | Abcam | Cat# ab107159 | |
| Antibody | Rabbit polyclonal anti-Hmgcs1 | Abcam | Cat# ab155787 | |
| Antibody | Rabbit polyclonal anti-Fdps | Abcam | Cat# ab153805 | |
| Antibody | Rabbit polyclonal anti-Lss | Abcam | Cat# ab80364 | |
| Antibody | Mouse monoclonal anti-RNA polymerase II CTD repeat YSPTSPS | Abcam | Cat# ab817 | |
| Antibody | Rabbit monoclonal anti-MAG | Cell Signal Technology | Cat# 9043 | |
| Antibody | Rabbit polyclonal anti-GFP | Cell Signal Technology | Cat# 2555 | |
| Antibody | Rabbit monoclonal anti-Pdgfrα | Cell Signal Technology | Cat# 3174 | |
| Antibody | Mouse monoclonal anti-Ki67 | Cell Signal Technology | Cat# 9449 | |
| Antibody | Rabbit polyclonal anti-Aspa | Millipore | Cat# ABN1698 | |
| Antibody | Rabbit polyclonal anti-Olig2 | Millipore | Cat# AB9610 | |
| Antibody | Mouse monoclonal anti-NeuN | Millipore | Cat# MAB377 | |
| Antibody | Mouse monoclonal anti-Apc (CC-1) | Millipore | Cat# OP80 | |
| Antibody | Rabbit polyclonal anti-Gstpi | MBL International | Cat# 311 | |
| Antibody | Mouse monoclonal anti-Qki (Pan) | Sigma-Aldrich | Cat# SAB5201536 | |
| Antibody | Rabbit polyclonal anti-Hmgcs2 | Sigma-Aldrich | Cat# SAB2107997 | |
| Antibody | Rabbit polyclonal anti-Srebp2 | Sigma-Aldrich | Cat# HPA031962 | |
| Antibody | Mouse monoclonal anti–β-actin | Sigma-Aldrich | Cat# A5441 | |
| Antibody | Mouse monoclonal anti-Flag | Sigma-Aldrich | Cat# F1804 | |

*Continued on next page*

*Continued*

| Reagent type (species) or resource | Designation | Source or reference | Identifiers | Additional information |
|---|---|---|---|---|
| Antibody | Rabbit polyclonal anti-Hmgcr | Thermo Fisher Scientific | Cat# PA5-37367 | |
| Antibody | Rabbit polyclonal anti-Srebp2 | Cayman Chemical | Cat# 10007663 | |
| Antibody | Mouse monoclonal anti-Gfap | BD Biosciences | Cat# 556330 | |
| Antibody | Normal rabbit IgG | Santa Cruz Technology | Cat# sc-2027 | |
| Antibody | Rabbit polyclonal anti–Qki-5 | This paper | immunized with a short synthetic peptide (CGAV ATKVRRHDMRVHPY QRIVTADRAATGN) | |
| Antibody | Mouse monoclonal anti-AnkG | Sigma-Aldrich | MABN466 | |
| Antibody | Rabbit polyclonal anti–Sox9 | Sigma-Aldrich | AB5535 | |
| Antibody | Rabbit polyclonal anti-Caspr | Gift from Dr. Rasband lab | | |
| Antibody | Mouse monoclonal anti-PanNav | Gift from Dr. Rasband lab | (K58/35) | |
| Chemical compound | Tamoxifen | Sigma-Aldrich | Cat# T5648 | |
| Chemical compound | 4-hydroxytamoxifen | Sigma-Aldrich | Cat# H7904 | |
| Chemical compound | Poly-L-ornithine | Sigma-Aldrich | Cat# P3655 | |
| Chemical compound | 3,3′,5-Triiodo-L-thyronine | Sigma-Aldrich | Cat# T2877 | |
| Chemical compound | Anti-Flag M2 Magnetic Beads | Sigma-Aldrich | Cat# M8823 | |
| Chemical compound | Citrate buffer | Poly Scientific R&D Corp | Cat# s2506 | |
| Chemical compound | FluoroMyelin Red | Thermo Fisher Scientific | Cat# F34652 | |
| Chemical compound | Penicillin-Streptomycin | Thermo Fisher Scientific | Cat# 15140122 | |
| Recombinant protein | Laminin Mouse Protein, Natural | Thermo Fisher Scientific | Cat# 23017015 | |
| Chemical compound | Neurobasal Medium | Thermo Fisher Scientific | Cat# 21103049 | |
| Chemical compound | B-27 | Thermo Fisher Scientific | Cat# 17504044 | |
| Chemical compound | GlutaMAX Supplement | Thermo Fisher Scientific | Cat# 35050061 | |
| Chemical compound | Puromycin Dihydrochloride | Thermo Fisher Scientific | Cat# A1113802 | |
| Recombinant protein | Dynabeads Protein G | Thermo Fisher Scientific | Cat# 10004D | |
| Chemical compound | HardSet Antifade Mounting Medium with DAPI | Vector Laboratories | Cat# H1500 | |
| Chemical compound | NeuroCult Basal Medium | Stemcell Technologies | Cat# 05700 | |
| Chemical compound | NeuroCult Proliferation Supplement | Stemcell Technologies | Cat# 05701 | |
| Recombinant protein | Recombinant Human EGF | ProteinTech | Cat# AF-100-15 | |
| Recombinant protein | Recombinant Human FGF-basic (146 a.a.) | ProteinTech | Cat# 100-18C | |

*Continued on next page*

*Continued*

| Reagent type (species) or resource | Designation | Source or reference | Identifiers | Additional information |
|---|---|---|---|---|
| Chemical compound | DNase (RNase-free) | Qiagen | Cat# 79254 | |
| Chemical compound | cOmplete, Mini Protease Inhibitor Cocktail | Roche Diagnostics | Cat# 11836153001 | |
| Chemical compound | RNase A | Thermo Fisher Scientific | Cat# 12091021 | |
| Chemical compound | Proteinase K (PK) Solution | Promega | Cat# MC5005 | |
| Commercial assay | Neural Tissue Dissociation Kit | Miltenyi Biotec | Cat# 130-092-628 | |
| Commercial assay | In-Fusion HD Cloning Plus | Takara Bio | Cat# 638910 | |
| Commercial assay | RNeasy Mini Kit | Qiagen | Cat# 74104 | |
| Commercial assay | QIAquick PCR Purification Kit | Qiagen | Cat# 28104 | |
| Commercial assay | SuperScript III First-Strand Synthesis SuperMix for qRT-PCR | Thermo Fisher Scientific | Cat# 11752250 | |
| Commercial assay | SuperSignal West Pico PLUS Chemiluminescence System | Thermo Fisher Scientific | Cat# 34579 | |
| Commercial assay | iTaq Universal SYBR Green Supermix | Bio-rad Laboratories | Cat# 1725122 | |
| Commercial assay | DC Protein Assay Kit | Bio-rad Laboratories | Cat# 5000121 | |
| Commercial assay | Total Cholesterol Assay Kits | Cell Biolabs | Cat# STA-390 | |
| Commercial assay | KAPA Hyper Prep Kit | Kapa Biosystems | Cat# 004477 | |
| Commercial assay | In Situ Cell Death Detection Kit | Millipore Sigma | 11684795910 | |
| Deposited data | Gene expression profile | This paper | GEO: GSE145116 | |
| Deposited data | Gene expression profile | This paper | GEO: GSE145117 | |
| Deposited data | ChIP sequencing data | This paper | GEO: GSE144756 | |
| Deposited data | ChIP sequencing data | *Zhou et al., 2020* | GEO: GSE126577 | |
| Experimental model | Mouse: B6.(Cg)-Nestin-CreER$^{T2}$ | *Imayoshi et al., 2008* | N/A | |
| Experimental model | Mouse: B6.(Cg)-Plp1-CreER$^{T2}$ | The Jackson Laboratory | Stock No.: 005975 | |
| Experimental model | Mouse: B6.129-Rosa26-CreER$^{T2}$ | The Jackson Laboratory | Stock No.: 008463 | |
| Experimental model | Mouse: B6.129(Cg)-ROSA$^{mT/mG}$ | The Jackson Laboratory | Stock No.: 007676 | |
| Experimental model | Mouse: B6.(Cg)-*Qk-loxP* | *Shingu et al., 2017* | N/A | |
| Recombinant DNA reagent | pLKO-puro Flag-Srebp2 | Addgene | Cat# 32018 | |

*Continued on next page*

*Continued*

| Reagent type (species) or resource | Designation | Source or reference | Identifiers | Additional information |
|---|---|---|---|---|
| Recombinant DNA reagent | pLKO-puro 3X Flag-Srebp2 | This paper | N/A | |
| Software and algorithm | ImageJ | National Institutes of Health | https://imagej.nih.gov/ij/ | |
| Software and algorithm | HISAT2 | Johns Hopkins University | https://ccb.jhu.edu/software/hisat2/index.shtml | |
| Software and algorithm | StringTie | Johns Hopkins University | https://ccb.jhu.edu/software/stringtie/ | |
| Software and algorithm | DESeq2 | Bioconductor | http://bioconductor.org/packages/DESeq2/ | |
| Software and algorithm | IPA | Qiagen | https://www.qiagenbioinformatics.com/products/ingenuity-pathway-analysis/ | |
| Software and algorithm | Trim Galore | Babraham Institute | http://www.bioinformatics.babraham.ac.uk/projects/trim_galore/ | |
| Software and algorithm | Bowtie | Johns Hopkins University | http://bowtie-bio.sourceforge.net/tutorial.shtml | |
| Software and algorithm | SAMtools | *Li et al., 2009* | https://github.com/samtools/samtools | |
| Software and algorithm | MACS2 | *Feng et al., 2012* | https://github.com/taoliu/MACS/ | |
| Software and algorithm | deeptools | *Ramírez et al., 2016* | https://github.com/deeptools/deepTools | |
| Software and algorithm | ngsplot | *Shen et al., 2014* | https://github.com/shenlab-sinai/ngsplot | |
| Software and algorithm | Prism 8 | GraphPad Software | https://www.graphpad.com/scientific-software/prism/ | |
| Software and algorithm | HOMER | UCSD | http://homer.ucsd.edu/homer | |

## Mouse models

*Nestin-CreER*[T2] mice (C57BL/6) were gifts from R. Kageyama (Kyoto University, Kyoto, Japan) (*Imayoshi et al., 2008*). *Plp-CreER*[T2] mice (C57BL/6) (*Doerflinger et al., 2003*), *Rosa26-CreER*[T2] mice (C57BL/6) (*Ventura et al., 2007*), and *mTmG* mice (C57BL/6) (*Muzumdar et al., 2007*) were obtained from The Jackson Laboratory (Bar Harbor, ME). Conditional *Qk*-knockout mice with two *loxP* sequences flanking exon 2 of the *Qk* gene (*Qk*[L/L]) were generated by our group as described previously (*Shingu et al., 2017*). *Qk*[L/L] mice were crossed with *Nestin-CreER*[T2] transgenic mice, *Plp-CreER*[T2] transgenic mice, or *R26-CreER*[T2] mice, in which expression of tamoxifen-inducible Cre was under the control of the *Nestin* promoter, *Plp* promoter, or *Gt(ROSA)26Sor* promoter, respectively. Mice were group-housed at MD Anderson's animal facility under pathogen-free conditions, maintained under a 12 hr light-dark schedule, allowed free access to water and food, and monitored for signs of illness every other day. All mouse experiments were conducted according to the NIH guidelines, and protocols (IACUC Study #00001392-RN01) for mouse procedures were approved by the Institutional Animal Care and Use Committee of The University of Texas MD Anderson Cancer Center.

Beginning at P7, *Nestin-CreER*[T2];*Qk*[L/L] mice were injected subcutaneously with 20 μL of tamoxifen (10 mg/mL) (Sigma-Aldrich) for two consecutive days to induce the deletion of *Qk*. Littermates (*Nestin-CreER*[T2];*Qk*[L/+] mice, *Nestin-CreER*[T2];WT mice, or *Qk*[L/L] mice) of the same age and genetic background were also injected with tamoxifen and used as controls for further experiments. Beginning at P4, *Plp-CreER*[T2];*Qk*[L/L], *Plp-CreER*[T2];*Qk*[L/+] mice, *Plp-CreER*[T2];WT mice, and *Qk*[L/L] mice were

injected subcutaneously with 10 µL of tamoxifen (10 mg/mL) for two consecutive days. After tamoxifen injection, the experimental mice were monitored daily, and their neurological impairments were recorded for further plotting of a quaking phenotype-free survival curve. When the *Qk*-iCKO mice reached the clinical endpoint, they had severe paralysis, significant weight loss, and hunched posture and were near death. Thus, they were euthanized for humanistic care.

## Rotarod behavioral analysis

The tamoxifen-injected mice were trained for five trials at a constant 5 rpm using a rotarod apparatus (Harvard Bioscience) and then tested in a 120 s trial at a constant 5 rpm. The latency of mice of falling off the rotarod was recorded. If a mouse stayed on the rotarod for more than 120 s, it was recorded as 120 s. Three trials were tested for each mouse at intervals of at least 30 min. Statistical analysis was performed using the mean latency of mice falling off the rotarod of three trials.

## Tissue preparation and immunofluorescence

Mice were anesthetized with isoflurane and transcardially perfused with 20 mL of 4% paraformaldehyde in phosphate-buffered saline (PBS). Their brains and optic nerves were dissected, postfixed in 4% paraformaldehyde at 4°C for 2 days, dehydrated in 30% sucrose in PBS at 4°C until the tissues sank, and embedded in optimal cutting temperature compound. Alternatively, brains and optic nerves were postfixed in formalin at room temperature for 2 days and embedded in paraffin. Frozen sections of 10 µm and paraffin-embedded sections of 5 µm were quickly boiled in citrate buffer (Poly Scientific R&D Corp.) for heat-induced epitope retrieval. They were then permeabilized with 0.25% Triton for 10 min and blocked with 10% horse serum for 1 hr at room temperature. The following primary antibodies were used for staining of the sections overnight at 4°C: anti-MBP (SMI-94R) and anti-β-amyloid (SIG-39220) were from BioLegend; anti-PLP (ab105784), anti-Iba1 (ab107159), anti-Hmgcs1 (ab155787), and anti-Fdps (ab153805) were from Abcam; anti-MAG (#9043), anti-GFP (#2555), anti-Pdgfrα (#3174), and anti-Ki67 (#9449) were from Cell Signaling Technology; anti-Aspa (ABN1698), anti-Olig2 (AB9610), anti-NeuN (MAB377), and anti–CC-1 were from Merck Millipore; anti-Gstpi (311) was from MBL International; anti-Qki (SAB5201536), anti-AnkG (MABN466), anti-Sox9 (AB5535), and anti-Srebp2 (HPA031962) were from Sigma-Aldrich and Millipore Sigma; and anti-Gfap (556330) was from BD Biosciences; anti-Caspr and anti-PanNav (K58/35) were gifts from Matthew N. Rasband (Baylor College of Medicine, Houston, TX, USA). TUNEL positive cells were detected using In Situ Cell Death Detection Kit (11684795910) from Millipore Sigma. The sections were incubated with appropriate Alexa Fluor dye-conjugated secondary antibodies (Thermo Fisher Scientific) for 1 hr at room temperature. FluoroMyelin (Thermo Fisher Scientific) was stained directly onto rehydrated slides for 20 min at room temperature according to the manufacturer's instructions. The sections were mounted using a VECTASHIELD antifade mounting medium with DAPI. A Leica DMi8 microscope was used to visualize most of the stained sections, and a Leica DFC345 FX digital monochrome camera was used to obtain fluorescent images of them. The images in *Figure 2C* and *Figure 2—figure supplement 1A* were taken using a Nikon Upright Eclipse Ni-E microscope.

## Electron microscopy

Beginning at P7, *Nestin-CreER*^T2^;*Qk*^L/L^ mice (*n* = 3) and control mice (*n* = 3) were injected with 20 µL of tamoxifen (10 mg/mL) on two consecutive days. Also, beginning at P4, *Plp-CreER*^T2^;*Qk*^L/L^ mice (*n* = 3) and control mice (*n* = 5) were injected with 10 µL of tamoxifen (10 mg/mL) on two consecutive days. Two weeks later, these experimental mice were transcardially perfused with 2% paraformaldehyde. Their optic nerves were then postfixed in a cold PBS solution containing 3% glutaraldehyde and 2% paraformaldehyde at 4°C and processed at the MD Anderson High Resolution Electron Microscopy Facility. In brief, these fixed optic nerves were washed in 0.1 M sodium cacodylate buffer, treated with 0.1% Millipore-filtered cacodylate-buffered tannic acid, postfixed with 1% buffered osmium tetroxide for 30 min, and stained *en bloc* with 1% Millipore-filtered uranyl acetate. The samples were dehydrated in increasing concentrations of ethanol and then infiltrated with and embedded in LX-112 medium. The samples were polymerized in a 60°C oven for about 3 days. Ultrathin sections of the samples were cut using a Leica Ultracut microtome, stained with uranyl acetate and lead citrate in a Leica EM stainer, and examined using a JEM 1010 transmission electron microscope (JEOL USA) at an accelerating voltage of 80 kV. Digital images of the sections were

captured using an AMT imaging system (Advanced Microscopy Techniques). ImageJ software (National Institutes of Health) was used to measure the axonal calibers and diameters of myelinated fibers; the percentage of myelinated axons, g-ratio, axonal diameter, and density of axon in these optic nerves were quantified on the basis of these measurements.

## NSC isolation and oligodendrocyte differentiation

The entire brains of *Nestin-CreER*^T2;*Qk*^L/L mouse pups at P1 were dissected, sliced into small pieces, and dissociated enzymatically into single cells using Neural Tissue Dissociation Kits (Miltenyi Biotec) according to the manufacturer's instructions. The single-cell suspension was then maintained in NeuroCult Basal Medium (STEMCELL Technologies) containing NeuroCult Proliferation Supplement (STEMCELL Technologies), 20 ng/mL epidermal growth factor (ProteinTech), 10 ng/mL basic fibroblast growth factor (ProteinTech), 50 U/mL penicillin G (Thermo Fisher Scientific), and 50 μg/mL streptomycin (Thermo Fisher Scientific) in a humidified 37°C incubator with 5% $CO_2$. All the cell cultures were negative for mycoplasma infection. To knock out *Qk*, NSCs mentioned above were treated twice with 100 nM 4-hydroxytamoxifen (Sigma-Aldrich) at 2-day intervals.

To induce in vitro oligodendrocyte differentiation, NSCs were seeded onto culture dishes pre-coated with 20 μg/mL poly-L-ornithine (Sigma-Aldrich) and 10 μg/mL laminin (Thermo Fisher Scientific) at $2.5 \times 10^4$ cells/cm$^2$ and cultured in the NSC medium described above. Two days later, the medium was changed to Neurobasal Medium (Thermo Fisher Scientific) supplemented with B-27 (Thermo Fisher Scientific), 2 mM GlutaMAX-I (Thermo Fisher Scientific), 30 ng/mL 3,3',5-triiodo-L-thyronine (Sigma-Aldrich), 50 U/mL penicillin G, and 50 μg/mL streptomycin. The cells were cultured in differentiation medium for 3 days, and fresh medium was replaced every other day.

## Stable cells

The coding DNA sequence region of Srebf2 was amplified from pLKO-puro Flag-Srebp2 (*Peterson et al., 2011*) using PCR and engineered into a pcDNA vector containing 2X Flag to generate an insert of Srebp2 with 2X Flag at the N-terminus of Srebp2 (pcDNA-2X Flag-Srebp2). pLKO-puro Flag-Srebp2 containing 1X Flag at the N-terminus was cut using *Sal*I and *Not*I to remove Srebp2, and pcDNA-2X Flag-Srebp2 was fused with the cut vector to generate an Srebp2-expressing vector with 3X Flag at the N-terminus (pLKO-puro 3X Flag-Srebp2) using an In-Fusion HD Cloning Kit (Takara Bio). Lentiviruses were packaged in HEK293T cells and used to infect NSCs. The cells were then selected with puromycin (Thermo Fisher Scientific) for 1 week, and the viable cells were used for further experiments.

## RNA isolation and real-time qPCR

After tamoxifen administration at P7, *Nestin-CreER*^T2;*Qk*^L/L mice and control mice were sacrificed at P21 to dissect the corpus callosum tissues (*n* = 4/group). Total RNA was extracted from those corpus callosum tissues using an RNeasy Mini Kit (QIAGEN) following the manufacturer's instructions. Two micrograms of total RNA were used to generate cDNA using an SuperScript III First-Strand Synthesis SuperMix (Thermo Fisher Scientific) . Real-time qPCR was carried out using a 7500 Fast Real-Time PCR system (Applied Biosystems) with iTaq Universal SYBR Green Supermix (Bio-Rad Laboratories). The quantitative transcription levels of the genes involved in cholesterol biosynthesis were normalized to expression of *Actb* and calculated using the ΔΔCT method. A complete list of the sequences of the primer pairs used is shown in *Supplementary file 1*.

## RNA-seq and pathway enrichment analyses

*Rosa26-CreER*^T2;*Qk*^L/L mice (*n* = 3) and control mice (*n* = 5) at P1 were injected with 10 μL of tamoxifen (10 mg/mL) and killed at P7 to collect their brains. Beginning at P7, *Plp-CreER*^T2;*Qk*^L/L mice (*n* = 2) and control mice (*n* = 2) were injected with 20 μL of tamoxifen (10 mg/mL) on two consecutive days, and they were sacrificed at P30 to collect their brains. Total RNA was isolated from their brains using an RNeasy Mini Kit with DNase treatment (RNase-free; QIAGEN). RNA-seq was performed by the Illumina HiSeq/MiSeq sequencing service at the MD Anderson Advanced Technology Genomics Core. The procedure of stranded paired-end RNA-seq analysis was performed according to Pertea's protocol (*Pertea et al., 2016*). In brief, the paired end reads were aligned with the mouse reference genome (mm10) and mouse reference transcriptome (GENCODE vM8) using

HISAT2 software (*Kim et al., 2015*) with the parameter '–rna-strandness RF –dta –no-mixed –no-discordant.' Gene expression levels were then calculated using StringTie software (*Pertea et al., 2015*) with GENCODE vM8 annotations of basic chromosomes. After the read counts were obtained using prepDE.py (*Pertea et al., 2015*), analysis of differential expression was conducted using DESeq2 (*Love et al., 2014*). The threshold for identifying differentially expressed genes was set at 1.5-fold (*R26* cohort) or 1.2-fold (*Plp* cohort) with p values less than 0.05. IPA software (QIAGEN) (*Krämer et al., 2014*) was used to analyze the significantly affected canonical signaling pathways, molecular and cellular functions, and upstream regulators associated with differentially expressed genes after Qki depletion.

## Cholesterol measurement

The corpus callosum tissues in *Qk*-Nestin-iCKO mice (*n* = 6) and control mice (*n* = 5 mice) were homogenized in a lipid extraction buffer (chloroform:isopropanol:NP-40, 7.0:11.0:0.1). After centrifugation at 13,000 g at 4°C for 10 min, the supernatant was collected and air-dried to remove the organic solvent. Dried lipid was used to measure cholesterol levels using Total Cholesterol Assay Kits (Cell Biolabs).

## IP and immunoblotting

The differentiated oligodendrocytes were washed once with PBS and cross-linked with 1% formaldehyde for 10 min. After quenching with 0.125 M glycine, cells were washed twice with cold PBS and lysed in NP-40 buffer (50 mM Tris-HCl, pH 7.4, 150 mM NaCl, 5 mM EDTA, 0.05% NP-40 supplemented with freshly added protease inhibitors) for 30 min at 4°C. The whole-cell lysate was then sonicated using a Bioruptor Pico sonication device (Diagenode) for 60 cycles (30 s on, 30 s off) on high-power setting. After centrifugation at 13,000 g at 4°C for 15 min to remove insoluble debris, the supernatant was incubated with antibodies against Qki-5 (immunizing rabbit with a short synthetic peptide [CGAVATKVRRHDMRVHPYQRIVTADRAATGN]; GenScript), Srebp2 (10007663; Cayman Chemical), or normal rabbit immunoglobulin G at 4°C overnight. The IP system was further rotated in the presence of magnetic recombinant protein G-coated beads (Thermo Fisher Scientific) for 2 hr at 4°C. Alternatively, co-IP was performed as described above except using anti-Flag M2 magnetic beads (Sigma-Aldrich) for 4 hr, and co-IP using denatured anti-FLAG M2 magnetic beads which were boiled at 95°C for 30 min was served as a control. Bound beads were then washed three times in cold NP-40 buffer by inverting the tubes, boiled in sample buffer at 95°C for 20 min, and subjected to sodium dodecyl sulfate-polyacrylamide gel electrophoresis and immunoblotting.

Corpus callosum tissue lysates were prepared by homogenizing the tissue in NP-40 buffer at 4°C as described above and quantified using a DC protein assay (Bio-Rad Laboratories) prior to immunoblotting. Briefly, 20–40 µg of proteins was electrophoresed on 4–12% gradient sodium dodecyl sulfate-polyacrylamide gels and transferred to nitrocellulose membranes. Subsequently, the membranes were blocked with 5% milk at room temperature for 1 hr and incubated with anti-Hmgcs1 (ab155787; Abcam), anti-Hmgcs2 (SAB2107997; Sigma-Aldrich), anti-Hmgcr (PA5-37367; Thermo Fisher Scientific), anti-Lss (ab80364; Abcam), anti-Fdps (ab153805; Abcam), anti-Flag (F1804; Sigma-Aldrich), and anti-β-actin (A5441; Sigma-Aldrich) antibodies overnight at 4°C. Membranes were then incubated with appropriate horseradish peroxidase-conjugated secondary antibodies, and the bands were visualized using a SuperSignal enhanced chemiluminescence system (Thermo Fisher Scientific). Blots were processed and quantified using ImageJ software, and β-actin was used for normalization.

## ChIP-seq and ChIP-qPCR

The ChIP assays were performed as described previously (*Lan et al., 2007*). In brief, adherent differentiated oligodendrocytes were cross-linked with 1% formaldehyde at room temperature for 10 min and quenched by 0.125 M glycine. Then, the cross-linked cells were suspended in ChIP lysis buffer (50 mM Tris-HCl, pH 7.4, 500 mM NaCl, 2 mM EDTA, 1% Triton X-100, 0.1% sodium dodecyl sulfate, 0.1% sodium deoxycholate with freshly added protease inhibitors). Sheared chromatin from these cells was diluted in ChIP dilution buffer (50 mM Tris-HCl, pH 7.4, 100 mM NaCl, 2 mM EDTA, 1% Triton X-100, 0.1% sodium deoxycholate with freshly added protease inhibitors) at a ratio of 1:1 and then incubated with anti–Qki-5 (immunizing rabbit with a short synthetic peptide [CGAVATKVRRHD

MRVHPYQRIVTADRAATGN]; GenScript), anti-Srebp2 (10007663; Cayman Chemical), or anti-Pol II (ab817; Abcam) antibodies overnight at 4°C. After immobilization on prewashed protein G agarose beads (Thermo Fisher Scientific), the protein-DNA complexes were washed three times with high-salt buffer (50 mM HEPES, pH 7.4, 500 mM NaCl, 1 mM EDTA, 1% Triton X-100, 0.1% sodium deoxycholate, 0.1% sodium dodecyl sulfate with freshly added protease inhibitors), twice with low-salt buffer (10 mM Tris-HCl, pH 8.1, 250 mM LiCl, 1 mM EDTA, 0.5% NP-40, 0.5% sodium deoxycholate with freshly added protease inhibitors), and once with TE buffer (10 mM Tris-HCl, pH 8.0, 1 mM EDTA). Elution and reverse cross-linking of DNA were carried out using elution buffer (50 mM Tris-HCl, pH 8.0, 10 mM EDTA, 1% sodium dodecyl sulfate) at 65°C for 4 hr. After digestion with RNase A (Thermo Fisher Scientific) and proteinase K (Promega) for 1 hr at 55°C, DNA samples were purified using a PCR Purification Kit (QIAGEN). Library preparation was performed using a KAPA HyperPrep Kit (Kapa Biosystems) according to the manufacturer's instructions and sequenced using an Illumina HiSeq X Ten system (Jiangxi Haplox Clinical Lab Cen, Ltd). FASTQ data were trimmed using Trim Galore (v0.4.4_dev) and mapped to the mouse genome (mm10 version) using Bowtie2 (v2.3.4.1) (*Langmead and Salzberg, 2012*) with the parameters '−1 R1.fastq −2 R2.fastq -X 1000', then the PCR duplication was removed by SAMtools (v1.8) (*Li et al., 2009*). Peaks were identified by MACS2 (v2.1.2) with the parameters 'macs2 callpeak -f BAMPE -g mm -q 0.05 -t ChIP.bam -n NAME -c INPUT.bam' (*Feng et al., 2012*). Bedgraph files were generated by deeptools (v3.3.0) (*Ramírez et al., 2016*) and uploaded to UCSC browser for visualization. Signal plots and heatmaps were generated using ngsplot (*Shen et al., 2014*). ChIP-seq was normalized by total reads. Motifs enriched in Qki-5 peaks were identified by HOMER (v4.10.1) with the following parameters: findMotifsGenome.pl peaklist.bed mm10 –size given –len 6,8,10,12,14 –mis 2 (*Heinz et al., 2010*). IPA software (*Krämer et al., 2014*) was used to analyze canonical signaling pathways enriched in genes whose promoters were co-occupied by Qki-5, Srebp2, and Pol II; overlapping genes among Qki-5-bound genes in freshly isolated mouse oligodendrocytes according to ChIP-seq and significantly downregulated genes in *Qk*-Plp-iCKO mice according to RNA-seq; overlapping genes among Qki-5-bound genes in differentiated oligodendrocytes according to ChIP-seq and significantly downregulated genes in *Qk*-Plp-iCKO mice according to RNA-seq; Srebp2-bound genes from Srebp2 ChIP-seq. ChIP-qPCR was performed using a 7500 Fast Real-Time PCR system (Applied Biosystems) with iTaq Universal SYBR Green Supermix (Bio-Rad Laboratories). All qPCRs were performed in triplicate, and a complete list of the sequences of the primers used is shown in *Supplementary file 1*.

## Statistics and reproducibility

All statistical analyses were performed using Prism 8 (GraphPad Software). No statistical methods were used for predetermining the sample size. The sample size was based on experimental feasibility, sample availability, and the number of necessary to obtain definitive results. The number of animals in each experiment is described in the corresponding figure legends. Numerical results are presented as means, with error bars representing standard deviation (s.d.). For comparison of two groups, a two-tailed, unpaired Student's $t$ test was used. To compare three or more groups, one-way analysis of variance (ANOVA) with Tukey's multiple comparisons test was conducted. Animal survival durations were analyzed using the log-rank test. Data distribution was assumed to be normal but has not been formally tested. All values of $p < 0.05$ were considered significant. No randomization or blinding events occurred during the experiments.

## Acknowledgements

We thank Shan Jiang, Kun Zhao, and Yanping Cao for mouse husbandry and care and all members of the Hu laboratory for helpful discussions. We thank Kenneth Dunner, Jr. for performing electron microscopy studies. We also thank Scientific Publications, Research Medical Library at MD Anderson for editorial assistance. This investigation was supported in part by grants from National Cancer Institute (R37CA214800) and the Cancer Prevention and Research Institute of Texas (RP120348 and RP170002). This research was supported by the University Cancer Foundation via the Institutional Research Grant program at the University of Texas MD Anderson Cancer Center. JH is supported by The University of Texas Rising STARs Award, the Sidney Kimmel Scholar Award, the Sontag Foundation Distinguished Scientist Award, and the Brockman Foundation. SS is supported by the Russell

and Diana Hawkins Family Foundation Discovery Fellowship, Sam Taub and Beatrice Burton Endowed Fellowship in Vision Disease, and Roberta M and Jean M Worsham Endowed Fellowship.

## Additional information

### Funding

| Funder | Grant reference number | Author |
|---|---|---|
| NCI | R37CA214800 | Jian Hu |
| University of Texas MD Anderson Cancer Center | startup | Jian Hu |
| Cancer Prevention and Research Institute of Texas | RP120348 | Jian Hu |
| Cancer Prevention and Research Institute of Texas | RP170002 | Jian Hu |

The funders had no role in study design, data collection and interpretation, or the decision to submit the work for publication.

### Author contributions

Xin Zhou, Seula Shin, Conceptualization, Data curation, Formal analysis, Validation, Visualization, Methodology, Writing - original draft, Writing - review and editing; Chenxi He, Formal analysis, Visualization, Methodology, Writing - review and editing; Qiang Zhang, Jiangong Ren, Congxin Dai, Takashi Shingu, Resources; Matthew N Rasband, Methodology; Rocío I Zorrilla-Veloz, Diagram illustration; Liang Yuan, Genotyping; Yunfei Wang, Software, Formal analysis; Yiwen Chen, Fei Lan, Resources, Formal analysis, Writing - review and editing; Jian Hu, Conceptualization, Supervision, Funding acquisition, Investigation, Writing - review and editing

### Author ORCIDs

Xin Zhou https://orcid.org/0000-0002-9362-8272
Seula Shin https://orcid.org/0000-0002-3593-5901
Matthew N Rasband http://orcid.org/0000-0001-8184-2477
Jian Hu https://orcid.org/0000-0001-9760-2013

### Ethics

Animal experimentation: All mouse experiments were conducted in accordance with protocols approved by the MD Anderson Institutional Animal Care and Use Committee. (IACUC Study #00001392-RN01).

### Decision letter and Author response

Decision letter https://doi.org/10.7554/eLife.60467.sa1
Author response https://doi.org/10.7554/eLife.60467.sa2

## Additional files

### Supplementary files

• Supplementary file 1. A complete list of the sequences of the primer pairs used in this study.
• Transparent reporting form

### Data availability

Sequencing data have been deposited in GEO under accession codes GSE145116, GSE145117 and GSE144756.

The following datasets were generated:

| Author(s) | Year | Dataset title | Dataset URL | Database and Identifier |
|---|---|---|---|---|
| Zhou X, Shin S, He C, Zhang Q, Ren J, Dai C, Shingu T, Yuan L, Wang Y, Chen Y, Lan F, Hu J | 2021 | RNA-seq-1 | https://www.ncbi.nlm.nih.gov/geo/query/acc.cgi?acc=GSE145116 | NCBI Gene Expression Omnibus, GSE145116 |
| Zhou X, Shin S, He C, Zhang Q, Ren J, Dai C, Shingu T, Yuan L, Wang Y, Chen Y, Lan F, Hu J | 2021 | RNA-seq-2 | https://www.ncbi.nlm.nih.gov/geo/query/acc.cgi?acc=GSE145117 | NCBI Gene Expression Omnibus, GSE145117 |
| Zhou X, Shin S, He C, Zhang Q, Ren J, Dai C, Shingu T, Yuan L, Wang Y, Chen Y, Lan F, Hu J | 2021 | Genome-wide maps of Qki-5, Srebp2, and Pol II in oligodendrocyte | https://www.ncbi.nlm.nih.gov/geo/query/acc.cgi?acc=GSE144756 | NCBI Gene Expression Omnibus, GSE144756 |

The following previously published dataset was used:

| Author(s) | Year | Dataset title | Dataset URL | Database and Identifier |
|---|---|---|---|---|
| Zhou X, He C, Ren J, Dai C, Stevens SR, Wang Q, Zamler D, Shingu T, Yuan L, Chandregowda CR, Wang Y, Ravikumar V, Rao A, Zhou F, Zheng H, Rasband MN, Chen Y, Lan F, Heimberger AB, Segal BM, Hu J | 2020 | Genome-wide maps of Qki-5 and PPARb in mouse oligodendrocytes | https://www.ncbi.nlm.nih.gov/geo/query/acc.cgi?acc=GSE126577 | NCBI Gene Expression Omnibus, GSE126577 |

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
