## [Decision Letter]

**Acceptance summary:**

Cholesterol is essential for myelin membrane assembly in the central nervous system, yet the mechanisms for temporal control of cholesterol biosynthesis during oligodendrocyte differentiation remain largely unknown. This study describes the role of Qki, commonly known as an RNA-binding protein, in controlling cholesterol biosynthesis in oligodendrocytes by acting as a transcriptional co-activator. These findings establish a previously unrecognized function of Qki in controlling cholesterol metabolism for temporal regulation of myelinogensis in developing central nervous system.

**Decision letter after peer review:**

Thank you for submitting your article "Qki regulates myelinogenesis through Srebp2-dependent cholesterol biosynthesis" for consideration by *eLife*. Your article has been reviewed by 3 peer reviewers, including Jian Xu as the Reviewing Editor and Reviewer #1, and the evaluation has been overseen by Marianne Bronner as the Senior Editor. The following individual involved in review of your submission has agreed to reveal their identity: Richard Q. Lu (Reviewer #3).

The reviewers have discussed the reviews with one another and the Reviewing Editor has drafted this decision to help you prepare a revised submission.

The reviewers found the study of significant interest by reporting a new mechanism linking Qki, cholesterol biosynthesis, myelinogenesis, and Srebp2-mediated transcriptional activation in oligodendrocytes. The experiments were well executed and the data supported the conclusions. Other main strengths of this work are the use of multiple in vivo genetic models and orthogonal approaches for studying cellular and molecular changes upon temporal Qki depletion, thus the technical rigor of this study was very high. Moreover, the studies uncovered a new function for Qki in regulating myelinogenesis by acting as a transcriptional co-activator for Srebp2 in developing brain. These findings provide new insights into the Qki regulation of myelinogenesis. The reviewers also had a number of comments to improve the manuscript, including the analysis of Qki KO on oligodendrocyte and astrocyte development, myelin structure and organization, and additional discussion points below. We have also included the separate review comments from the reviewers.

The reviewers had different views on whether the work represents a significant advance in light of the recent publication from the authors' group reporting a role for Qki in myelin maintenance by functioning as a coactivator of PPARb-RXRa-mediated lipid metabolism (Zhou et al., 2020 JCI 130:2220). Therefore, besides addressing the following comments, it is important to include a section to describe the major differences in the current work and published studies, and the major conceptual advances of the current study. A diagram may be more effective in conveying these messages.

Essential revisions:

1. Regarding the Nestin-cKO studies. Is Qk expressed in astrocytes? The P7 Tm treatment will likely also eliminate Qk in astrocytes. Accordingly, the authors should confirm that astrocyte development is not impacted in this mouse line.

2. A more in-depth analysis of the effect on OPC development is needed. Specifically, a Cre-dependent reporter would be needed to confirm the impact of Qki loss on oligodendrocyte development in the tamoxifen-induced iCKO mice. The proliferation and survival of OPCs and oligodendrocyte in iCKO lines should be included to make sure that the differentiation defects are not due to the progenitor deficits.

3. The introduction touches on tangential topics of psychiatric disorders and white matter, etc.

This manuscript does not endeavor to examine psychiatric disorders, thus introducing this topic is not necessary as its distracting and links between the current studies and this topic are non-existent.

4. Does myelin properly form in these mice? What about Node formation and AIS? The authors have nice EM data, but fail to show whether the myelin organization is affected and whether the axons themselves are impacted. If nodes and AIS forms, then it might be worthwhile to do a time course analysis to decipher when these structures breakdown.

*Reviewer #1:*

In this manuscript, Hu and colleagues characterized the role of Qki in oligodendrocyte differentiation and myelinogenesis by conditional and inducible inactivation of Qki in neural stem cells (NSCs) or oligodendrocyte precursor cells (OPCs) in neonatal mice during the myelin-forming period of postnatal brain development. The authors found that Qki depletion in NSCs or OPCs impaired cholesterol biosynthesis and myelinogenesis without affecting differentiation into Aspa+Gstpi+ myelinating oligodendrocytes. They next performed RNA-seq-based transcriptomics and ChIP-seq-based chromatin occupancy studies, and identified Qki-5 as a transcriptional co-activator of Srebp2 to control the expression of genes involved in cholesterol biogenesis in oligodendrocytes. Qki depletion impaired Srebp2-mediated transcriptional activation of genes required for cholesterol biosynthesis, thus establishing a new function of Qki as a transcriptional co-activator beyond its known role as an RNA-binding protein.

Overall, this is an important and well-executed study reporting a new and interesting mechanism linking Qki, cholesterol biosynthesis, myelinogenesis, and Srebp2-mediated transcriptional activation in oligodendrocytes. The experiments related to the phenotypic analysis of Qki conditional KO in NSCs and OPCs, and the mechanistic studies of Qki in developing oligodendrocytes were appropriately designed, and the results were carefully analyzed. Another main strengths of this work are the use of multiple in vivo genetic models and orthogonal approaches for studying cellular and molecular changes upon Qki depletion, thus the technical rigor of this study was excellent. Moreover, the detailed functional and mechanistic studies uncovered a new function for Qki in regulating myelinogenesis by acting as a transcriptional co-activator for Srebp2 in developing brain. The manuscript was well written, and the results were carefully interpreted and discussed.

*Reviewer #2:*

In this paper by Hu and colleagues, the role of the Quaking gene in developmental myelination is explored. Using temporal-lineage specific mouse genetics, they show that loss of Qk results in severe defects in myelination. Mechanistic studies show that cholesterol biosynthesis pathways are dysregulted and that Qk collaboratively regulates subsets of these gens with Srebp2, a known transcriptional regulator of cholesterol associated genes.

Overall, this paper is well executed and the data support the conclusions. The main issue with this manuscript is that the authors just published a very nice paper in JCI showing essentially the same role of Qk in the adult. The only real difference between the published JCI paper and this current manuscript is the timing of the administration of tamoxifen: in the JCI paper it was added at 8w of age, here its added at P7. One could imagine the P7 treatment being included as part of the original analysis conducted for the published JCI paper. The fact that Qk plays an important role in lipid/cholesterol in myelin has already been shown, making the findings contained within this paper incremental and not of interest to a broad audience.

For these reasons, I do not support publication of this manuscript in *eLife*, as it's more appropriate for specialized journals that target glial biologists and those interested in demyelinating disorders. Below are a few comments meant to improve a well-executed, but incremental paper:

1. Regarding the Nestin-cKO studies. Is Qk expressed in astrocytes? The P7 Tm treatment will likely also eliminate Qk in astrocytes. Accordingly, the authors should confirm that astrocyte development is not impacted in this mouse line.

2. The fact that OPCs appear unaffected further supports the notion that Qk plays a role in myelin biosynthesis via the mechanisms described in their JCI paper (and in this paper). If they want to uncoupled their developmental studies, from their adult studies, a more in-depth analysis of OPC biology and transcriptome might help.

3. The introduction touches on tangential topics of psychiatric disorders and white matter, etc.

This manuscript does not endeavor to examine psychiatric disorders, thus introducing this topic is not necessary as its distracting and links between the current studies and this topic are non-existent.

4. One myelin related question. Does myelin ever properly form in these mice? Have the authors looked at Node formation? What about the AIS? The authors have nice EM data, but fail to show whether the myelin organization is affected and whether the axons themselves are impacted. If nodes and AIS forms, then it might be worthwhile to do a time course analysis to decipher when these structures breakdown.

*Reviewer #3:*

QKI is essential for CNS myelination and myelin maintenance, however, the underlying mechanisms remain unclear. In this study, Dr. Hu and colleagues defined the of QKI at different stages during oligodendrocyte development. They show that the loss of Qki results in defects in OPC differentiation and myelination but not OPC formation. They further show that Qki5 can interact with a crucial cholesterol biosynthesis regulator Srebp2 and that QKi5 and Srebp2 cooperate to regulate cholesterol biosynthesis genes. The findings that link QKi5 function to Srebp2 and cholesterol biosynthesis is interesting and provide a new insight into the Qki regulation of myelinogenesis. Following concerns need to be addressed to strengthen their conclusions.

1. A cre-dependent reporter would need to confirm the impact of Qki loss on oligodendrocyte development in the tamoxifen-induced iCKO mice.

2. The authors should examine the proliferation and survival of OPCs and oligodendrocyte in iCKO lines to make sure that the differentiation defects are not due to the progenitor deficits.

3. Qk undergoes alternative splicing to express Qki-5, Qki-6, and Qki-7. Qk deletion may also reduce other isoforms besides Qki-5. Could Qki-6 and Qki-7 have a role in OPC differentiation and myelination too?

4. Previous study suggested that nuclear-localized QKI-5 predominantly represses myelination while the cytoplasmic QKI-6 and QKI-7 are essential for promoting OL maturation and myelination (PMID: 20956316). The present study suggests a positive role QKI-5 in cholesterol synthesis and CNS myelination. Could the authors comment on these different observations on the QKI-5 function in OL maturation?

5. Could the authors show a significant reduction of Pol II or Srebp2 target occupancy upon Qki deletion with statistic tests in Figure 7D,E? Could Qki have a role in global gene transcription in addition to cholesterol biosynthesis genes?

6. Is there any evidence that Qki-5 functions as a co-activator of Srebp2 for cholesterol biosynthesis? Since the authors reported Qki-5 interact with PPARb-RXRa for lipid metabolism for myelination, could the authors define how Qki-5 coordinates PPARb-RXRa or Srebp2 for myelination?

---

## [Author Response]

The reviewers found the study of significant interest by reporting a new mechanism linking Qki, cholesterol biosynthesis, myelinogenesis, and Srebp2-mediated transcriptional activation in oligodendrocytes. The experiments were well executed and the data supported the conclusions. Other main strengths of this work are the use of multiple in vivo genetic models and orthogonal approaches for studying cellular and molecular changes upon temporal Qki depletion, thus the technical rigor of this study was very high. Moreover, the studies uncovered a new function for Qki in regulating myelinogenesis by acting as a transcriptional co-activator for Srebp2 in developing brain. These findings provide new insights into the Qki regulation of myelinogenesis. The reviewers also had a number of comments to improve the manuscript, including the analysis of Qki KO on oligodendrocyte and astrocyte development, myelin structure and organization, and additional discussion points that are grouped into essential revisions and minor comments below. We have also included the separate review comments from the reviewers.The reviewers had different views on whether the work represents a significant advance in light of the recent publication from the authors' group reporting a role for Qki in myelin maintenance by functioning as a coactivator of PPARb-RXRa-mediated lipid metabolism (Zhou et al., 2020 JCI 130:2220). Therefore, besides addressing the following comments, it is important to include a section to describe the major differences in the current work and published studies, and the major conceptual advances of the current study. A diagram may be more effective in conveying these messages.

We thank all the editors and reviewers for their enthusiastic and constructive feedback. On the basis of the comments from the editors and reviewers, we have performed 11 experiments and added 20 new figure panels in the revised manuscript. Lastly, we included the following points to clarify the major differences between the current work and the published studies (in particular, Zhou et al., JCI, 2020) and summarize the major conceptual advances of the current study.

1. The focus of the current study is to investigate the transcriptional regulatory mechanism for de novo production of the major myelin lipid –cholesterol– in oligodendrocytes during early developmental stage, whereas the JCI paper (Zhou et al., 2020) focuses on the transcriptional regulation of biosynthesis of the unsaturated fatty acids and very-long-chain fatty acids (which are the main components of phospholipids, sphingolipid, and glycolipids) in oligodendrocytes of adult mice. Although both cholesterol and fatty acids are important lipids in lipid-rich myelin membrane, cholesterol specifically functions as a chaperone in proper sorting of newly synthesized hydrophobic myelin membrane proteins such as PLP during the developmental process (Simons and Trotter, 2007). Besides, cholesterol is a major component of microdomain on the myelin membrane that facilitates the proper localization of myelin proteins and the translation of MBP mRNA (Gielen et al., 2006; Hughes and Appel, 2016; Simons et al., 2000). In the current study, we showed that significant reduction in cholesterol level induced by Qki depletion leads to defect in myelin protein localization observed by patchy MBP staining and impairment of co-localization of myelin proteins such as MBP, PLP, and MAG (Figure 3A, B).

Importantly, the defects in myelin protein assembly in *Qk*Nestin-iCKO mice upon Qki loss were not observed in the JCI paper. In summary, compared to phospholipids, sphingolipid, and glycolipids, cholesterol plays a distinct role in facilitating myelin membrane formation; our current study is the first one to demonstrate that Qki regulates cholesterol biosynthesis in oligodendrocytes.

2. Molecular machineries that are involved in transcriptional regulation of cholesterol biosynthesis are different from those in fatty acid metabolism. One of the major differences is the mechanisms through which the major transcription factors are regulated. For example, PGC1α is a critical regulator of PPARβ-RXRα, but it has no direct impact on Srebp2. In addition, Srebp2 is post-translationally processed and translocalized to the nucleus with the help of scaffold proteins such as Scap and Insig, which can sense the availability of cellular cholesterol. In contrast, the main regulatory mechanism of PPARβ-RXRα heterodimer is through ligand binding, which in turn facilitates the interaction with transcription co-factors and activates transcription of fatty acid metabolic genes. The biosynthesis of cholesterol and fatty acids have to be coordinated, yet few co-factors have been identified to be shared by Srebp2 and PPARβ-RXRα. Our JCI paper and the current study discovered that Qki is actually a cofactor for both Srebp2 and PPARβ-RXRα, thereby functions as a major regulator that coordinates the biosynthesis of cholesterol and fatty acids in myelination.

3. Although both cholesterol ((Saher et al., 2005) and the current study) and fatty acids (Dimas et al., 2019) are essential for de novo myelination, the stability of these types of lipids are quite different. In the JCI paper, we discovered that fatty acids in mature myelin undergo fast turnover, and Qki-5/PPARβ/RXRα is the major regulator for fatty acid metabolism in mature myelin. However, in the JCI paper, we didn’t identify cholesterol metabolism as a pathway that is regulated by Qki in the adult mice, and the reason is that unlike fatty acids, cholesterol is actually quite stable in mature myelin with a half-life of more than 5 years in human and more than 1 year in mouse (Ando et al., 2003; Russell et al., 2009; Smith, 1968), so the cholesterol metabolic genes are generally suppressed in adult oligodendrocytes because the constant biosynthesis of cholesterol is not needed due to its slow turnover rate. Therefore, developmental stage is the only period in which cholesterol metabolism can be effectively studied in oligodendrocytes, and this is why only the current study but not the JCI paper could discover Qki as a major regulator of cholesterol biosynthesis.

4. It has been known that reduced cholesterol level in *qk*^v^ mice is secondary to impairment of oligodendrocyte differentiation and maturation during development (Baumann et al., 1968; Singh et al., 1971). However, in the current study, we uncovered a previously uncharacterized function of Qki in controlling transcription of the genes involved in cholesterol biosynthesis without affecting the differentiation of Aspa^+^Gstpi^+^ myelinating oligodendrocytes, and we propose that cholesterol metabolism controlled by Qki-5 is a determinant for proper function of myelinating oligodendrocytes, and ultimately for temporal control of CNS myelinogenesis.

Essential revisions:1. Regarding the Nestin-cKO studies. Is Qk expressed in astrocytes? The P7 Tm treatment will likely also eliminate Qk in astrocytes. Accordingly, the authors should confirm that astrocyte development is not impacted in this mouse line.

Qki is moderately expressed in both nucleus and cytosol in GFAP^+^ astrocytes of P21 mice, while GFAP^-^ cells (predicted to be oligodendrocyte lineage cells) show stronger Qki expression (>2 fold higher than that in GFAP^+^ astrocytes) with predominant localization in the nucleus (Author response image 1).

**Author response image 1. sa2fig1:** Qki is moderately expressed in GFAP+ astrocytes. Representative images and quantification of immunofluorescent staining of Qki and GFAP in the cortex/corpus callosum/hippocampus tissues in WT mice at P21. Arrow: Qki+GFAP+ cells. Arrow head: Qki+GFAP- cells. CTX: cortex. CC: corpus callosum. HC: hippocampus. Scale bars, 100 μm.

We provide the following evidence to show that astrocyte development is not impacted by Qki depletion. First, a Cre-dependent mTmG reporter was used for in-depth analysis of the effect of Qki depletion on astrocyte development. In the P21 *NestinCreER^T2^*;*mTmG* mice, the percentage of GFP^+^GFAP^+^ cells among the total GFAP^+^ astrocytes is 9.48% (Author response image 2, B); In the P21 *Qk*-Nestin-iCKO;*mTmG* mice, the percentage of GFP^+^GFAP^+^ cells (indicating Qki-depleted astrocytes) among the total GFAP^+^ astrocytes is 11.59%, which is similar as that in control (Author response image 2, B), indicating that depletion of Qki in *Qk*-Nestin-iCKO mice does not affect the astrocyte development. Second, *Sox9* has been shown to be a specific and potent marker for astrocyte by Dr. Ben Deneen and others (Laug et al., 2019). So, to reinforce our finding, we further co-labeled GFAP^+^ astrocytes with *Sox9* (Sun et al., 2017).

**Author response image 2. sa2fig2:** GFP is expressed in a small subpopulation of GFAP+ astrocytes, and Qki loss does not alter GFAP expression. (**A**) Representative images of immunofluorescent staining of GFP and GFAP in the corpus callosum tissues in *Qk*- Nestin-iCKO and control mice two weeks after tamoxifen injection. Arrowhead: GFAP+GFP-cells. Arrow: GFAP+GFP+ cells. Scale bars, 100 μm. (**B**) Quantification of GFAP+GFP+ cells in *Qk*-Nestin-iCKO (n = 4) and control (n = 4) mice two weeks after tamoxifen injection shown in A. (**C**) Quantification of relative GFAP expression in GFAP+GFP+ cells from *Qk*-Nestin-iCKO (n = 4) and control (n = 4) mice two weeks after tamoxifen injection shown in A.

Similarly, *Sox9*^+^GFAP^+^GFP^+^ cells only constituted a small population of total *Sox9*^+^GFAP^+^ astrocytes in both *Qk*-Nestin-iCKO;*mTmG* mice (15.92%) and control *Nestin*-*CreER^T2^*;*mTmG* mice (16.22%) (New Figure 2—figure supplement 2B). Collectively, these data suggested that majority of GFAP^+^ astrocytes are developed prior to P7 and therefore are not targeted by *Nestin*-*CreER*^T2^ inducible system with P7 tamoxifen treatment. Previous studies have shown that astrocyte precursor cells are derived from neural stem cells around E16-E18 (Ge et al., 2012), and astrocyte precursors and astrocytes are actively expanding in number and significantly decrease Nestin expression during the first postnatal week (Cahoy et al., 2008; Clavreul et al., 2019), supporting our observation that the impact of Qki depletion under the control of *Nestin*-*CreER*^T2^ promoter at P7 on astrocyte population is quite limited. We have included this result in text line “241-248” in the revised manuscript.

In addition, the recruitment of GFP^+^GFAP^+^ cells to the dysmyelinating regions and enhancement of GFAP expression in these cells were not affected upon Qki depletion (Author response image 2), further suggesting that Qki loss does not alter the astrocyte activity.

Taken together, our data indicated that: (1) P7 tamoxifen injection in the *NestinCreER^T2^*mice only leads to expression of Cre activity in around 10% of astrocytes which are newly differentiated from neural stem cells after P7; (2) depletion of Qki in these astrocytes does not lead to impairment of their development at P21; and (3) depletion of Qki in these astrocytes does not lead to impairment of their activity at P21. Lastly, the dysmyelinating phenotypes observed in the *Qk*-Nestin-iCKO mice were further validated in the *Qk*-Plp-iCKO mice.

2. A more in-depth analysis of the effect on OPC development is needed. Specifically, a Cre-dependent reporter would be needed to confirm the impact of Qki loss on oligodendrocyte development in the tamoxifen-induced iCKO mice.

In our original manuscript, we found that the number of Pdgfrα^+^ OPCs in the developing cortex tissues in *Qk*-Nestin-iCKO mice was slightly higher than that in control mice (Figure 2A), probably due to a compensatory increase in the proliferation of OPCs in response to hypomyelination. In consistence with our finding, a previous study also showed that the number of Pdgfrα^+^ OPCs in *QKI^FL/FL;Olig2–Cre^* mice was slightly higher than that in control mice (Darbelli et al., 2016). Notably, 92.6% of Pdgfrα^+^ OPCs in *Qk*Nestin-iCKO mice lacked expression of Qki, indicating that Qki does not affect the generation or survival of OPCs (Figure 2A).

The activity of the *Plp1* promoter in the CNS of early neonatal mice is restricted to a subset of OPCs poised to differentiate into myelinating oligodendrocytes (Guo et al., 2010). To further determine the effect of Qki on OPC development, we crossed the mice bearing the *mTmG* reporter line (Muzumdar et al., 2007) with *Qk*-Plp-iCKO mice or control mice. The mTmG reporter, in which expression of cell membrane-localized tdTomato (mT) is replaced by cell membrane-localized EGFP (mG) in Cre recombinase-expressing cells, enabled us to trace newly formed oligodendroglial lineage cells (after tamoxifen injection) including a subset of OPCs according to the GFP signals. On the basis of this Cre-dependent reporter, we confirmed that *Plp1*-*CreER*^T2^; mTmG cohort labels OPC population as indicated by the Pdgfrα^+^GFP^+^ double positive cells (New Figure 4—figure supplement 1F). More importantly, Qki loss did not alter the number of Pdgfrα^+^GFP^+^ cells (New Figure 4—figure supplement 1F), suggesting that the development and survival of OPC population was not altered upon Qki depletion. We have included this result in text line “316-319” in the revised manuscript.

The proliferation and survival of OPCs and oligodendrocyte in iCKO lines should be included to make sure that the differentiation defects are not due to the progenitor deficits.

No alteration in proliferation was observed upon Qki depletion in OPC population (Pdgfrα^+^ cells) or oligodendroglial lineage cells (Olig2^+^ cells) co-labeled by proliferating marker, Ki67 (New Figure 4—figure supplement 2A, B) between *Qk*-PlpiCKO and control. In addition, comparable numbers of TUNEL positive cells (which are very few) were found between *Qk*-Nestin-iCKO and control (New Figure 2—figure supplement 1C). These data suggest that depletion of Qki does not affect the proliferation and survival of OPC, and the defect in differentiation of Olig2^+^Aspa^-^Gstpi^-^ oligodendroglial lineage cells was not due to impaired OPC development. We have included this result in text line “320-325” in the revised manuscript.

3. The introduction touches on tangential topics of psychiatric disorders and white matter, etc.This manuscript does not endeavor to examine psychiatric disorders, thus introducing this topic is not necessary as its distracting and links between the current studies and this topic are non-existent.

We have removed this part in the Introduction section as the reviewer suggested.

4. Does myelin properly form in these mice?

To decipher whether myelin was properly formed in *Qk*-Nestin-iCKO mice, a time course analysis of MBP expression was performed in the corpus callosum. As shown in New Figure 1—figure supplement 1C, Qki depletion slowed down the myelin formation during brain development observed from P12 (5 days after tamoxifen injection) to P21 (14 days after tamoxifen injection), during which myelin is initiated and actively generated, ultimately resulting in failure of proper myelination formation and motor deficit at P21 (Figure 1C, Video 1). We have included this result in text line “161-166” in the revised manuscript. In addition to the corpus callosum tissues, hypomyelination was observed in optic nerves of *Qk*-Nestin-iCKO mice (Figure 1G) at P21, suggesting that Qki depletion leads to global abnormalities in the CNS myelin formation.

What about Node formation and AIS? The authors have nice EM data, but fail to show whether the myelin organization is affected and whether the axons themselves are impacted.

To ask if the formation of node of Ranvier is affected by Qki depletion, we focused on examining the node formation in the optic nerve, where rapid and robust myelination occurs during development. By labeling the nodes using antibodies against paranodal protein (Caspr) and nodal proteins (AnkG or PanNav), we found that Qki depletion leads to defect in intact node formation at P21 (14 days after P7 tamoxifen injection) (New Figure 1—figure supplement 1D). Consistently, paranodal defects were also observed in *QKI^FL/FL;Olig2–Cre^* mice in a previous study (Darbelli et al., J. Neurosci, 2016). We have included this result in text line “166-169” in the revised manuscript.

In addition to the node formation, we further asked if Qki loss impacts on the axon initial segment (AIS) structure. We found that the length of AIS labeled by AnkG at the proximal axon adjacent to cell body (NeuN) was not altered in the cortex region of Nestin-Qki-iCKO mice compared to control mice as shown in New Figure 1—figure supplement 1K-M, suggesting that generation of action potential was not affected upon Qki depletion. We have included this result in text line “187-191” in the revised manuscript.

Although myelin formation was severely affected, we did not observe the alteration of thickness and number of axons (New Figure 1—figure supplement 1I, J). In addition, no axonal damage was found, as shown in Figure 1I.

If nodes and AIS forms, then it might be worthwhile to do a time course analysis to decipher when these structures breakdown.

As node formation was severely affected by Qki depletion at P21, we further examined when this occurred during early myelin development through a time course analysis. Previous studies showed that clustering of ion channels at the nodes requires proper myelination (Rasband and Peles, 2020; Rasband et al., 1999). As myelination is rapidly formed at its peak from P14 in the optic nerve (Mayoral et al., 2018), we monitored earlier times (P14 and P17) to ask if myelin defect induced by Qki depletion affects de novo formation of nodes. Defect in node formation was observed as early as P14, which cannot be overcome at P21 (New Figure 1—figure supplement 1E, F), and this observation is in line with the defect in myelination (New Figure 1—figure supplement 1C). Specifically, total number of nodes (including both intact and the incomplete nodes) in the optic nerve was decreased upon Qki depletion (New Figure 1figure supplement 1G). Importantly, the percentage of intact nodes among the total nodes was also significantly reduced with Qki depletion (New Figure 1—figure supplement 1H). Notably, Qki depletion at P14 showed streaks of Caspr^+^ signals instead of clustering/localization of Caspr, a component of the axoglial junctions indicating the proper paranode formation (New Figure 1—figure supplement 1E, F). As it was previously reported that paranode formation precedes node clustering in the CNS (Rasband et al., 1999; Zhang et al., 2020), our observation suggests that failure of paranode formation due to defect in myelination upon Qki depletion ultimately leads to failure of node formation during the critical time of myelin development. We have included this result in text line “169-182” in the revised manuscript.

It has been shown that Qki depletion in oligodendrocyte lineage exhibits ultrastructural paranodal defects caused by reduced expression of neurofascin 155, an axoglial junctional protein (Darbelli et al., 2016). Additionally, myelin-specific lipid called galactolipids (GalC) has been studies to be essential for proper CNS node formation (Dupree et al., 1998), implying the importance of myelin lipid in node formation. Our study further sheds light on the understanding of the importance of myelin lipid metabolism (particularly cholesterol) regulated by Qki for proper myelination and formation of node of Ranvier.